# Children born after assisted reproduction more commonly carry a mitochondrial genotype associating with low birthweight

Joke Mertens[1], Florence Belva[2], Aafke P. A. van Montfoort[3], Marius Regin[1], Filippo Zambelli[4], Sara Seneca[1,2], Edouard Couvreu de Deckersberg[1], Maryse Bonduelle[2], Herman Tournaye[5,6], Katrien Stouffs[1,2], Kurt Barbé[7], Hubert J. M. Smeets [8,9], Hilde Van de Velde[5,10], Karen Sermon[1], Christophe Blockeel[5,11] & Claudia Spits [1] ✉

Children conceived through assisted reproductive technologies (ART) have an elevated risk of lower birthweight, yet the underlying cause remains unclear. Our study explores mitochondrial DNA (mtDNA) variants as contributors to birthweight differences by impacting mitochondrial function during prenatal development. We deep-sequenced the mtDNA of 451 ART and spontaneously conceived (SC) individuals, 157 mother-child pairs and 113 individual oocytes from either natural menstrual cycles or after ovarian stimulation (OS) and find that ART individuals carried a different mtDNA genotype than SC individuals, with more de novo non-synonymous variants. These variants, along with rRNA variants, correlate with lower birthweight percentiles, independent of conception mode. Their higher occurrence in ART individuals stems from de novo mutagenesis associated with maternal aging and OS-induced oocyte cohort size. Future research will establish the long-term health consequences of these changes and how these findings will impact the clinical practice and patient counselling in the future.

The birth of Louise Brown in 1978 heralded the start of a revolution in reproductive medicine that has made parenthood possible for couples previously considered untreatably infertile. Since then, assisted reproductive technologies (ART) have gone through considerable developments and their use has increased exponentially, with over 8 million children born worldwide. Over the past decades, virtually all steps of the process have undergone continuous improvement.

Nevertheless, concerns about the safety of these procedures regarding the health of the children born after ART have always been present, and the systematic studies of large cohorts of children over many years have revealed that these individuals have indeed an adverse health outcome as compared to those born after spontaneous conception (SC). There is by now much evidence that ART children are at increased risk of being born with a congenital abnormality and small for

[1]Research Group Reproduction and Genetics, Faculty of Medicine and Pharmacy, Vrije Universiteit Brussel, Brussels, Belgium. [2]Center for Medical Genetics, UZ Brussel, Brussels, Belgium. [3]Department of Obstetrics & Gynaecology, GROW School for Oncology and Developmental Biology, Maastricht University Medical Center, Maastricht, The Netherlands. [4]Basic Research Laboratory, Eugin Group, Barcelona, Spain. [5]Brussels IVF, Center for Reproductive Medicine, UZ Brussel, Brussels, Belgium. [6]Research Group Biology of the Testis, Faculty of Medicine, Vrije Universiteit Brussel, Brussels, Belgium. [7]Interfaculty Center Data Processing & Statistics, Vrije Universiteit Brussel, Brussels, Belgium. [8]Department of Toxicogenomics, Maastricht University, Maastricht, The Netherlands. [9]MHeNs School Institute for Mental Health and Neuroscience, GROW Institute for Oncology and Developmental Biology, Maastricht University, Maastricht, The Netherlands. [10]Research Group Reproduction and Immunology, Faculty of Medicine and Pharmacy, Vrije Universiteit Brussel, Brussels, Belgium. [11]Department of Obstetrics and Gynaecology, School of Medicine, University of Zagreb, Šalata 3, Zagreb 10000, Croatia. ✉ e-mail: claudia.spits@vub.be

gestational age[1,2], with ART singletons having an overall 2- to 3-fold increased risk of adverse perinatal outcomes[2–4]. In contrast, children from a frozen embryo transfer have a higher risk of being born large for gestational age[5], which associates with difficulties at delivery, childhood malignancies and cardio-metabolic disorders[6,7]. Long-term follow-up studies show increasing evidence that these children are at risk for abnormal hormonal and cardio-metabolic profiles later in life[8–14].

Over the years a large body of research has been devoted to the identification of the causes and molecular basis of these adverse effects in ART children, based on the knowledge that suboptimal early development conditions have a significant impact on the future health of the individual[15]. Three factors have mainly been studied in association with the health of the ART children, namely maternal subfertility, ovarian stimulation (OS) and embryo culture systems. However, the sometimes-contradicting results of different studies make it difficult to parse out the exact role of each factor in the phenotype of the children. First, the adverse health outcomes seen in ART children have been associated with the subfertility of their mothers[4,16–18]. In line with this, SC children born to subfertile mothers also have an increased risk of adverse perinatal outcomes[18–20], suggesting a major role for the maternal (genetic) background. Second, the procedure of OS, used in ART to increase the cohort of oocytes available in one menstrual cycle has been shown to impact birthweight[4]. It has been suggested that the maternal uterine environment under the influence of OS acts as an independent contributor to the risk for lower birthweight in children born following fresh embryo transfer, in contrast to frozen embryo transfer[21]. However, comparative studies of children born after OS with SC children from subfertile parents found no significant differences[22,23]. Third, the in vitro manipulation of embryos, including freezing and thawing, and particularly the use of different culture media has been shown to have a significant impact on birthweight[24–26]. Because ART centres throughout the world use different media, and these have been evolving over time, establishing the exact impact and the involved underlying factors in the media on the health of the ART child remains challenging[27,28].

From a molecular point of view, the vast majority of studies have focused on the search for ART-induced epigenetic changes, both in individuals and in placental samples. Although a significant number of animal studies have demonstrated a link between epigenetic alterations and different aspects of ART, studies in human did not provide conclusive results[29]. While some studies have found differences in the epigenetic landscape of ART and SC children[30,31], these were usually within normal range of variation. Many other researchers have not identified consistent methylation abnormalities in the genome of ART children[32–36], leading to results that are globally difficult to harmonize and with no association to the health outcomes of ART children. Lastly, a recent study showed that in vitro fertilization does not increase the incidence of chromosomal abnormalities in human fetal and placental lineages, and did not find a link between the presence of these abnormalities and birthweight[37].

In this study, we hypothesized that the differences in birthweight between ART and SC children are caused by mitochondrial DNA (mtDNA) variants. In the general population, low birthweight is associated with metabolic abnormalities[38], which in adulthood have been linked to mitochondrial dysfunction[39]. Furthermore, inherited mtDNA defects can predispose to obesity and insulin resistance[40], low birthweight is common in individuals born with mtDNA disease[41,42] and non-disease-associated mtDNA variation has been previously found to be associated with body composition and weight[43–50]. As mentioned above, female subfertility and OS strongly associate with the risk for low birthweight in ART children[4,51], and both may be linked to mitochondrial function as oxidative phosphorylation plays a crucial role during gametogenesis, embryonic development as well as in female subfertility[52–58]. We hypothesized that subfertile women more frequently carry mtDNA variants that negatively affect mitochondrial function and contribute to their subfertility. Furthermore, work in animal models suggests that OS increases the frequency of mtDNA deletions in the oocytes[59,60] and affects mitochondrial function[61]. Therefore, we postulated that maternal transmission of mtDNA genotypes, that might be related to subfertility, and de novo mutagenesis during OS could result in adverse perinatal outcomes in children carrying these mtDNA genotypes.

To test if the mtDNA genotypes of ART and SC children differ, if these differences are associated with birthweight percentile and to determine the origin of these potential differences, we studied the mitochondrial genome of 270 ART and 181 SC children, 157 ART and SC mother-child pairs and 113 oocytes donated in both natural menstrual cycles and after OS from the same donors by massively parallel sequencing.

## Results

### Individuals born after ART display a different mitochondrial DNA variant genotype than their spontaneously conceived peers

First, we sequenced the full mitochondrial genome of 270 ART and 181 SC children (details on the participants and the source of DNA can be found in Table 1). During data processing, we 1037 bp that comprised regions that were prone to PCR and sequencing errors (Fig. 1a).

First, the data was controlled for differences in haplogroup distribution. No significant differences were found in haplogroup and subhaplogroup distribution across ART and SC individuals (Fig. 1b, c, and Supplementary Tables 1 and 2). Next to the haplogroup variants, most individuals carried additional homoplasmic variants, which were categorized according to their location in the mitochondrial genome and their impact on the amino acid sequence: hypervariable region (HV), non-coding, origin of replication on the heavy strand (OHR), termination-associated sequence (TAS), rRNA, tRNA, synonymous (Syn) and non-synonymous (Non-syn) protein-coding variants. The proportion of individuals carrying homoplasmic variants was similar in both groups, as well as the location of the variants and their potential pathogenicity (Fig. 1d and Supplementary Table 3). One SC individual carried a proven pathogenic tRNA-coding variant at homoplasmic level (mt.14674 T > C, associated with Reversible Infantile Respiratory Chain Deficiency[62]).

In total, 430 heteroplasmic variants were identified, 66.0% of which were unique to one individual (an overview is shown in Fig. 1e). Since heteroplasmic variants may differ across different tissues and increase with age[63–66], we first controlled the dataset for the use of different tissues as source of DNA and different ages at sampling. We found no significant differences in the totals of the different heteroplasmic loads per individual in regard to their sample's source DNA or age at sampling (Fig. 1f, g). Next, we categorized the heteroplasmic variants per location, as done for the homoplasmic variants. 60.4% of ART and 61.9% of SC individuals carried up to 5 heteroplasmic variants per person, with no significant differences in their distribution (Fig. 1h, Supplementary Table 4). There were also no statistically significant differences in the distributions of variants in non-coding regions, tRNA, rRNA and synonymous and non-synonymous variants, although ART individuals slightly more often carried non-synonymous protein-coding variants (Fig. 1h, Supplementary Table 4).

There were no differences in the distribution of the variants according to their location, type of nucleotide substitution and potential pathogenicity (Fig. 1i, Supplementary Tables 5 and 6). Four individuals (3 ART and 1 SC individuals) carried a proven pathogenic heteroplasmic tRNA-coding variant, including the variants m.3243 A > G and m.5521 G > A (both associated with the MELAS/MERRF syndrome[67,68]), and m.5703 G > A (associated with mitochondrial myopathy and MERRF[69]). To factor in the heteroplasmic load and to reduce the data complexity due to the diverse location of the variants,

we added up the loads of the heteroplasmic variants per individual, categorized per location (HV, tRNA, etc). This resulted in a matrix of "sums of heteroplasmic loads" per individual, per region. ART and SC individuals did not differ in the mean sum of heteroplasmic loads per location in their mtDNA (Fig. 1j, Supplementary Table 7). Next, we used an orthogonally rotated exploratory factor analysis on the sum of the heteroplasmic loads of each location in the mtDNA per individual. This further reduced the complexity of the data to four factors; the component matrix can be found in Fig. 1k. These factors reflect covariance between types of mtDNA variants, as represented by the hypercube in

Fig. 1l. We extracted the scores for each factor for all samples and used these to further explore diversity in heteroplasmic variant composition across the two groups. ART individuals were found to have significantly higher scores in factor 2, driven by the presence of protein-coding and rRNA variants and lack of HV, OHR, TAS and non-coding variants, as compared to SC children. Factors 1, 3 and 4 were not different between the two groups (Fig. 1m). This suggests that individuals born after ART tend to more frequently carry heteroplasmic variants in the protein-coding and/or rRNA regions, in combination with fewer heteroplasmic variants in the other regions compared to their SC peers. This is in line with the (non-statistically significant) trend of ART individuals having a higher number of non-synonymous variants (Fig. 1h) and a lower mean heteroplasmic load in the HV region (ART: 4.55%, SC: 7.02%, Fig. 1j), and illustrates how the factor analysis extracts a significant interaction between variables that a univariate analysis would not detect.

## Table 1 | Characteristics of the ART and SC individuals included in this study

| | ART (N = 270) % (N) | SC (N = 181) | P-value |
|---|---|---|---|
| **Source material** | | | |
| Blood[1] | 38.1% (103) | 35.9% (65) | |
| Newborns | 17.0% (46) | 0.0% (0) | |
| Young adults | 21.1% (57) | 35.9% (65) | |
| Placenta[2] | 24.4% (66) | 28.2% (51) | |
| Buccal cells[3] & Saliva[2] | 37.4% (101) | 35.9% (65) | |
| Children | 37.4% (101) | 2.8% (5) | |
| Young adults | 0.0% (0) | 33.1% (60) | |
| Mother-child pairs | 67 | 90 | |
| **Clinical parameters** | | | |
| Mean maternal age[4] ± SD* | 32.5 ± 3.7 (211) | 30.1 ± 4.4 (173) | <0.0001[a] |
| Maternal BMI > 25* | 40.2% (86/214) | 25.6% (42/164) | 0.0031[b] |
| Birthweight* | (224) | (164) | |
| Mean (grams) ± SD | 3377.9 ± 536.0 | 3394.3 ± 500.7 | 0.7596[a] |
| <10th percentile | 7.0% (19) | 5.0% (9) | 0.2407[b] |
| <25th percentile | 15.2% (41) | 14.4% (26) | 0.4992[b] |
| Preterm birth (<37 weeks)* | 5.8% (13/224) | 6.1% (10/164) | 1.0000[b] |
| Sex* | (224) | (181) | |
| (Male–Female) | 41.5% (112) – 41.5% (112) | 44.8% (81) – 55.2% (100) | 0.3176[b] |
| Culture medium | 224 | NA | |
| Vitrolife® G3 | 19.3% (52) | NA | |
| Vitrolife® G5 | 24.4% (66) | NA | |
| Cook® | 18.1% (49) | NA | |
| UZB | 21.1% (57) | NA | |
| Unknown | 17.0% (46) | NA | |
| Pregnancy hypertension* | 4.1% (8/193) | 1.9% (3/159) | 0.3571[b] |
| Gestational diabetes* | 1.0% (2/193) | 1.3% (2/158) | 1.0000[b] |
| Primiparity* | 75.2% (167/222) | 47.2% (68/144) | <0.0001[b] |
| Smoking during pregnancy* | 5.5% (12/220) | 7.7% (13/169) | 0.4086[b] |

The donors were recruited in different centres: [1]UZ Brussel, Belgium, [2]Maastricht University Medical Center, The Netherlands and [3]Faculty of Medicine and Pharmacy, Vrije Universiteit Brussel, Belgium. [4]Age in years.
*ART* assisted reproductive technologies, *SC* spontaneously conceived, *SD* standard deviation, *BMI* body mass index.
*Only a subset of samples was provided with clinical data such as maternal age, maternal BMI (body mass index), birthweight (categorized in percentiles), gestational age, sex, culture medium, complications during pregnancy and parity. UZB medium was an in-house medium made in the UZ Brussel. All individuals are singletons from fresh embryo transfers. Some characteristics were not applicable (NA) for some samples. *N* indicates the number of samples for which this data was available. Two-sided Statistical tests were performed using the student t test ([a]) or the Fisher's exact test ([b]).

### Differences in the mtDNA variant profile correlate with birth-weight percentiles

We investigated the association between the birthweight of 388 individuals and their mtDNA profile (164 SC and 224 ART individuals). The birthweight was adjusted for gestational age and sex and categorized as under or above the 10th percentile (<P10), where <P10 is considered small for gestational age, and under or above the 25th percentile (<P25).

First, we queried our dataset for variants that have been reported to associate with birthweight[43–50]. Our dataset did not contain individuals carrying the variants reported in refs. 46–48. In regard to the variant m.16189T, which has been associated with thinness at birth[43,44,47], 57 individuals in our study cohort carry this variant as a homoplasmy (often as part of their haplogroup) or as a heteroplasmy. However, it was not associated with a birthweight under the 10th or 25th percentile in our study cohort (Data not shown, Chi-square test, $p = 1$ and $p = 0.851$). These results did not change in terms of significance if considering the variant only when it is present outside of the haplogroup or as a heteroplasmy. Furthermore, haplogroup T, which has been associated with obesity[45,50], showed a trend to being under-represented in individuals under the 10th and 25th birthweight percentile ($p = 0.094$ and $p = 0.04$, respectively, however, these values are not significant after Bonferroni correction for multiple testing, Fig. 2a, b and Supplementary Tables 8 and S9). There was no association between other haplogroups and birthweight percentile nor with the presence of homoplasmic variants (Fig. 2c, Supplementary Tables 10 and 11), regardless of mode of conception.

Conversely, SC individuals who were <P10 or <P25 had significantly higher scores in factor 2 of the heteroplasmic variants analysis than children born at >P10 or >P25 (Fig. 2d). In line with this, as factor 2 is driven by rRNA and non-synonymous variants, SC individuals with a lower birthweight more frequently carried non-synonymous and rRNA variants, while no differences were observed for the other variants (<P10: 66.6% and >P10: 25.2%, Fisher's exact test, $p = 0.014$, <P25: 53.8% and >P25: 22.5%, Fisher's exact test, $p = 0.003$, Supplementary Tables 12 and 13). SC children <P10 and <P25 also carried higher heteroplasmic loads of rRNA variants (Fig. 2e), of non-synonymous variants (Fig. 2e) and of non-synonymous and rRNA variants combined (<P10 $p = 0.011$, <P25 $p = 0.001$, Fig. 2e). Strikingly, these differences were not observed in the ART individuals (Supplementary Tables 14 and 15). To establish which other variables were potentially affecting the results, we first stratified the samples by sex (Fig. 2f, g). This stratification resulted in the <P25 SC females having a significantly higher incidence of non-synonymous and rRNA variants ($p = 0.012$), while SC males <P25 and SC females and males <P10 showed higher incidences with p-values of 0.098, 0.087 and 0.093, respectively. In the ART group, only <P25 ART females had modestly but not significantly higher incidences of non-synonymous and rRNA

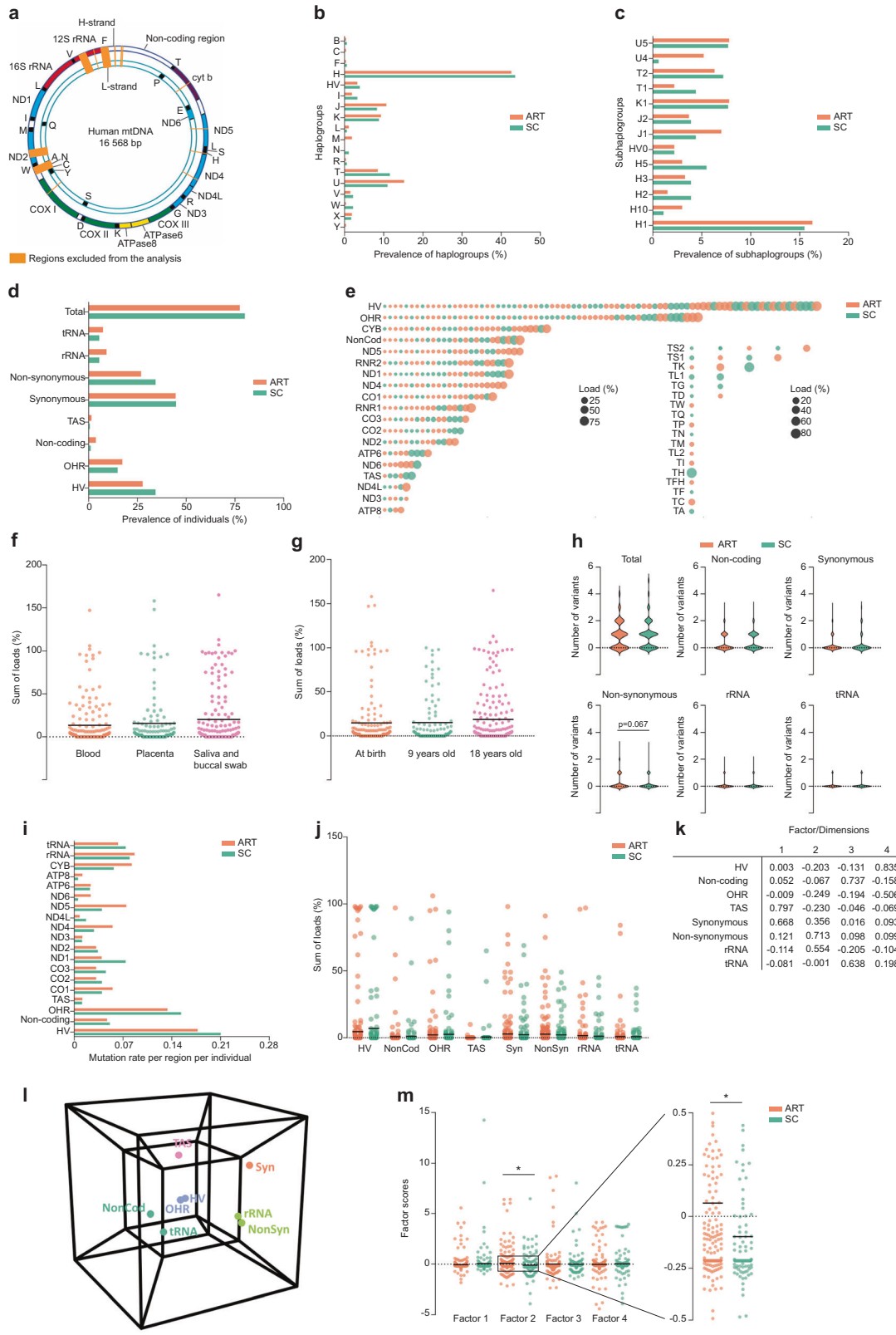

variants (*p* = 0.2075, Fig. 2f, g). Another potential variable in our cohort are the various embryo culture media to which the embryos were exposed. Our study group partially overlaps with that of a study showing that embryos that have been grown in Cook® medium have lower birthweights, and that male embryos are more sensitive to this effect[70]. We observe a similar trend in the present study, where for individuals that were subjected to Cook® and UZB medium (i.e. an in-

house medium made in the UZ Brussel) males are more frequently born with a birthweight <P25 than females (Fig. 2h, 25.5% in males vs 10.7% in females, *p* = 0.0745). Furthermore, only one of these <P25 male ART individuals (5%) carried non-synonymous or rRNA variants, as compared to 53.8% of the <P25 SC individuals (Fig. 2h, *p* = 0.001). In contrast, individuals that were exposed to Vitrolife® culture medium carried non-synonymous and rRNA variants in 50% of cases, similarly

**Fig. 1 | ART Individuals carry a different mitochondrial DNA genotype than their spontaneously conceived peers. a** Schematic overview of the mtDNA indicating in orange the regions that were excluded from the analysis. Adapted with permission from ref. 112, Copyright © 2009 International Union of Biochemistry and Molecular Biology, Inc. **b, c** Fisher's exact test shows no significant differences in the prevalence of the different haplogroups (**b**) and subhaplogroups (**c**) across ART and SC individuals. **d** Fisher's exact test shows no significant differences in the prevalence of ART and SC individuals with homoplasmic variants located in the different categories. Percentages do not add up to 100% because one individual can have more than one variant. **e** Overview of all heteroplasmic variants identified. Four of the tRNA genes were not represented in our dataset. Every bubble represents one variant and the size of the bubble is proportional to the variant's heteroplasmic load. **f, g** Sum of the heteroplasmic loads identified per sample, categorized according to the tissue of origin (**f**) and age at sampling (**g**).

Kruskal–Wallis Test shows no significant differences. Blood $n = 168$, Placenta $n = 117$, Saliva and Buccal Swab $n = 166$, at birth $n = 163$, 9 years old $n = 106$, 18 years old $n = 182$. **h** Number of variants identified per individual. Chi-square test shows no significant differences (non-synonymous variants, $p = 0.067$). **i** Mutation rate per region or protein-coding gene, per individual in ART and SC. **j** Sum of loads of the heteroplasmic variants found per individual and per location. Mann–Whitney U Test shows no significant differences. $n = 270$ ART, $n = 181$ SC. **k** Component matrix generated by the factor analysis. **l** Hypercube representing the covariance between the different categories of heteroplasmic variants. The colours indicate categories that co-variate in the same factor. **m** ART individuals more frequently carry higher scores for factor 2 (*Mann–Whitney U Test, $p = 0.021$). $n = 270$ ART, $n = 181$ SC. Horizontal bars in panels (**f**), (**g**), (**j**) and (**m**) represent the mean of the values in the scatter plot, each dot represents a sample. Source data are provided as a Source data file.

to SC individuals <P25 (Fig. 2h). These results suggest that the effect of Cook® or UZB medium exposure supersedes any effect of the mtDNA landscape. Therefore, in further analysis, we excluded ART individuals exposed to Cook® and UZB medium and we reanalysed all mtDNA-related factors for SC and ART individuals exposed to Vitrolife® medium only. There was no association between (sub)haplogroup and birthweight percentile (Fig. 3a, b and Supplementary Tables 16 and 17) nor with the presence of homoplasmic variants, with the exception of tRNA homoplasmies, that were more common in individuals <P10 (29.4% vs 6.4%, Fig. 3c, Supplementary Tables 18 and 19). When looking into the heteroplasmic variants, <P25 and <P10 individuals more frequently carried non-synonymous protein-coding variants and rRNA variants, but only reaching statistical significance for the non-synonymous variants in the <P25 individuals (Fig. 3d, Supplementary Tables 20 and 21). As in most cases the non-synonymous variants were not known to be associated with mitochondrial disease, we categorized them on whether there was further supporting evidence for their potential pathogenicity. Fifteen variants (18.75%) had a MutPred2 score over 0.61[71] and forty-six variants (57.7%) had a reported frequency under 0.002 in MITOMAP (mitomap.org[72]), a threshold that has been proposed as supporting evidence for pathogenicity[73]. This incidence of uncommon variants in our population is remarkably higher than that found in the MITOMAP database, where 19% of variants meet this threshold[73]. Taking the MutPred2 scores and the MITOMAP frequencies together, 50 of the 80 identified non-synonymous variants have supporting evidence for being potentially pathogenic (62.5%, Supplementary Data 1). Comparison of their incidence in individuals under and above P25 yielded no statistically significant differences, likely due to the limited sample size (21% in <P25, 14% in >P25, Fisher's exact test, $p = 0.270$), and we did not further factor this subdivision of the variants in the subsequent analysis. Combining non-synonymous and rRNA variants shows that 52.9% of <P10 and 52.1% of <P25 individuals carry at least one of such variants (vs >P10: 31.3% and >P25: 28.6%, Fig. 3e, Supplementary Tables 20 and 21). Additionally, SC and Vitrolife®-ART individuals with a birthweight <P10 and <P25 had overall higher cumulative heteroplasmic loads of non-synonymous and rRNA variants (Fig. 3f).

Next, we used binary logistic regression to study the effect of the different mtDNA-related variables on the birthweight, including the potential confounding factors listed in Table 1. We used two approaches: first, we included any mtDNA parameter that showed an association with birthweight percentile through univariate analysis with $p < 0.2$ in the model, with the exception of the presence of TAS homoplasmies because they were present in only 3 individuals of the cohort (Supplementary Data 2). Second, we tested a backward conditional approach, in which we introduced all variables listed in Supplementary Data 2. For individuals <P10, the following variables showed a statistically significant impact on birthweight: smoking during pregnancy, pregnancy hypertension, the presence of tRNA

homoplasmies outside of the person's haplogroup and the presence of haplogroup K1-associated variants (Table 2). These same factors were also found to significantly associate with birthweight, with the exception of the haplogroup K1-associated variants (Supplementary Table 22). In the case of individuals <P25, the logistic regression resulted in a model including haplogroups I, J and T, maternal age, the presence of tRNA homoplasmies outside of the person's haplogroup and the presence of heteroplasmic non-synonymous protein- and rRNA-coding variants. Only haplogroup I, maternal age and the presence of heteroplasmic non-synonymous and rRNA variants had a statistically significant impact in increasing the chances of being born <P25 (Table 2). The backward conditional model retained maternal age and the presence of heteroplasmic non-synonymous and rRNA variants as significantly associating with a birthweight <P25 (Supplementary Table 23). Although the logistic regression identified factors that had a statistically significant effect on the prediction of the birthweight percentile, the models generated by the regression were not effective in predicting if an individual would have a birthweight <P10 or <P25. In the <P10 cohort, only 1 out of the 13 individuals was correctly classified, whereas in the <P25 cohort, there were 4 individuals out of 42. We therefore developed an additional model based on the method of discriminant analysis. We built three models: the first using only the pregnancy-related factors (maternal age, smoking, pregnancy hypertension, maternal BMI > 25, primiparity and gestational diabetes), the second using only the mtDNA factors that associate with birthweight

**Table 2 | Binary logistic regression to predict a birthweight under the 10th or 25th percentile in SC individuals and ART individuals exposed to Vitrolife® culture medium**

|  | Exp(B) | 95% C.I. for Exp(B) | Significance |
|---|---|---|---|
| **P10 in SC and ART Vitrolife®** | | | |
| Smoking | 7.211 | 1.552–33.496 | 0.012 |
| Pregnancy hypertension | 14.045 | 1.185–166.410 | 0.036 |
| Haplogroup K1 | 4.952 | 1.075–22.813 | 0.040 |
| tRNA homoplasmies | 7.518 | 1.850–30.553 | 0.005 |
| Heteroplasmic non-synonymous and rRNA variants | 2.180 | 642–7.400 | 0.212 |
| **P25 in SC and ART Vitrolife®** | | | |
| Maternal age | 1.129 | 1.036–1.231 | 0.006 |
| Haplogroup I | 5.774 | 1.125–29.624 | 0.035 |
| Haplogroup J | 2.248 | 0.783–6.455 | 0.132 |
| Haplogroup T | 0.146 | 0.018–1.165 | 0.069 |
| tRNA homoplasmies | 2.405 | 0.796–7.264 | 0.120 |
| Heteroplasmic non-synonymous and rRNA variants | 2.974 | 1.439–6.2144 | 0.003 |

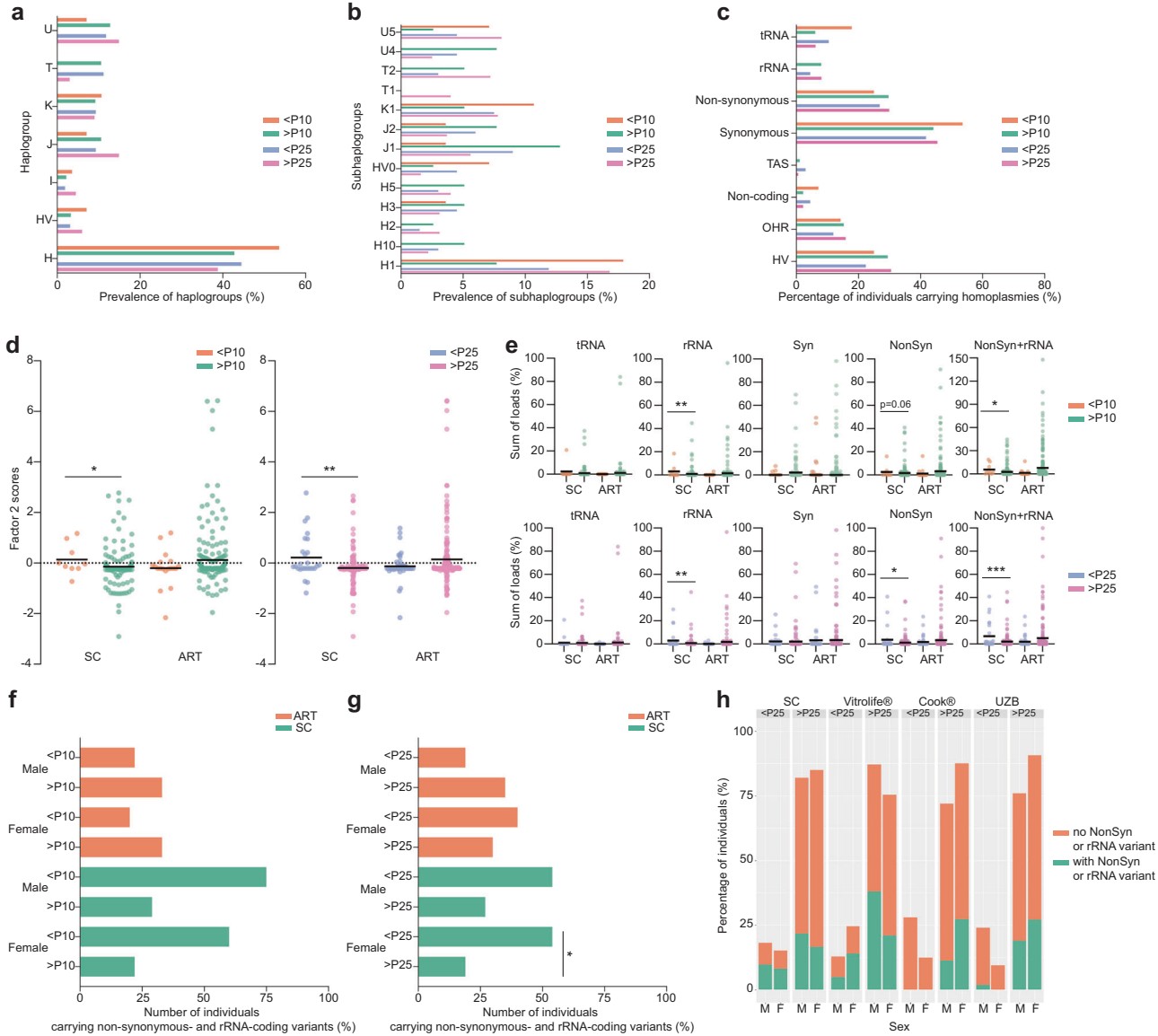

**Fig. 2 | Differences in mtDNA variant composition associate with birthweight percentile in SC individuals. a**, **b** No significant differences in the prevalence of the haplogroups and subhaplogroups across birthweight percentiles, two-sided Fisher's exact test with Bonferroni correction. **c** No significant differences in the percentages of individuals carrying homoplasmic variants in the different regions and categorized according to their birthweight percentile, two-sided Fisher's exact test with Bonferroni correction. **d** Factor 2 scores in ART and SC children categorized according to their birthweight percentile. two-sided Mann–Whitney U test, *$p = 0.04$ and **$p = 0.003$, $n$ ART < P10 = 19, $n$ ART > P10 = 205, $n$ SC < P10 = 9, $n$ SC > P10 = 155, $n$ ART < P25 = 41, $n$ ART > P25 = 183, $n$ SC < P25 = 26, $n$ SC > P25 = 138 individuals. **e** Heteroplasmic loads of tRNA, rRNA, and synonymous (Syn) and non-synonymous (Non-syn) protein-coding variants found in SC and ART individuals,

categorized to their birthweight percentile. Two-sided Man–Whitney U test, for rRNA variants <P10 **$p = 0.01$, <P25 **$p = 0.007$, for non-synonymous variants <P10 $p = 0.067$, <P25 *$p = 0.034$ and of non-synonymous and rRNA variants combined, * for <P10 $p = 0.011$, <P25 ***$p = 0.001$. **f**, **g** Number of ART and SC individuals, stratified for sex and birthweight percentile under or above P10 or P25, respectively, carrying non-synonymous and rRNA loci variants. <P25 SC females have a significantly higher incidence of non-synonymous and rRNA variants (* two-sided Fisher's exact test, $p = 0.012$). **h** Percentages of SC and ART individuals being under and above P25 and carrying non-synonymous and rRNA loci variants, stratified by sex, and by culture medium to which the embryos were exposed. Horizontal bars in panels (**d**), and (**e**) represent the mean of the values in the scatter plot, each dot represents a sample. Source data are provided as a Source data file.

percentile outcome with $p < 0.2$, and the third model combining both (Tables 3 and 4). The models to predict the <P10 and <P25 showed that, while the pregnancy-related factors on their own were very good predictors of the birthweight percentile, the addition of the mtDNA factors increased the ability of the model to correctly identify the individuals <P10 and <P25. For the <P10 model, the addition of the mtDNA factors increased the accuracy considerably for identifying the <P10 individuals from 40% to 70%. For <P25, the accuracy increased more modestly from 58.1% to 67.7%. Remarkably, the mtDNA factors on their own were more predictive for a birthweight <P25 than the

pregnancy-related factors (64.9% vs 62.5% correct classification, respectively).

## ART individuals more frequently show de novo non-synonymous variants than their spontaneously conceived peers

To investigate the origin of these differences in the mtDNA, we studied the mitochondrial genome of 67 ART and 90 SC mother-child pairs. We categorized a variant as "transmitted" when it was present in the mother and in the child, and as "de novo" when the variant was only present in the child. In total, we found 19 transmitted variants,

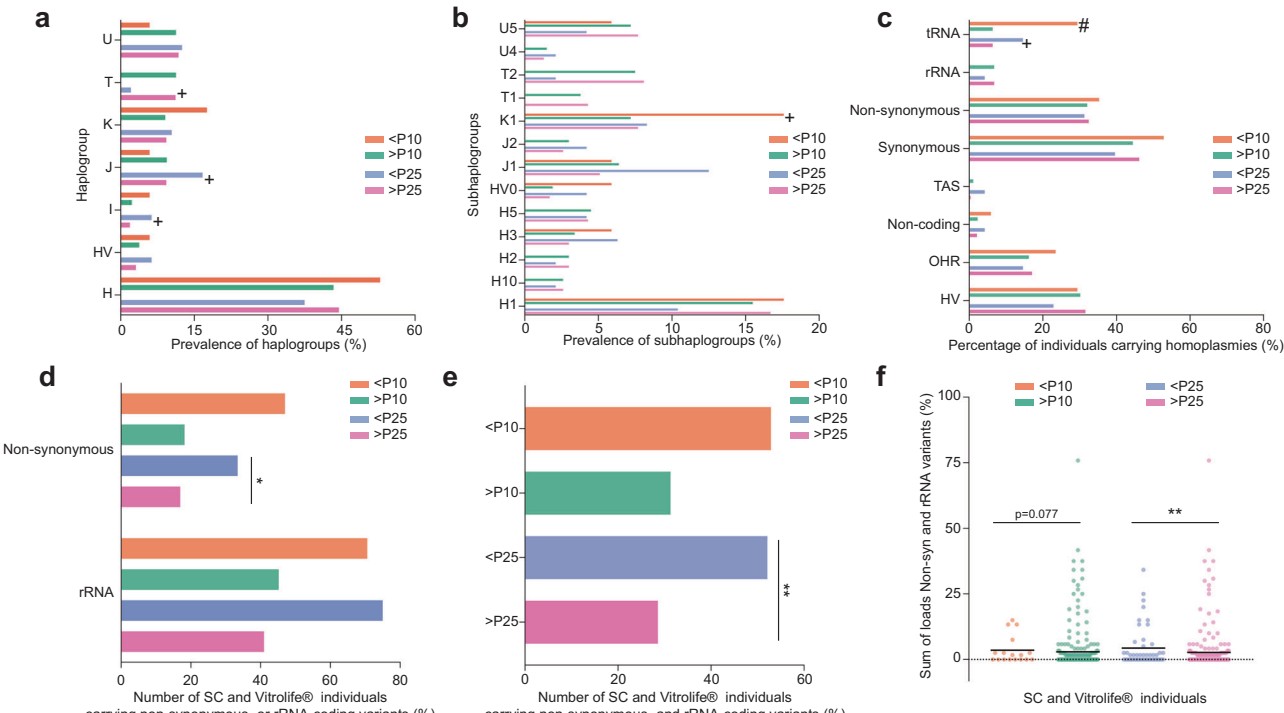

**Fig. 3 | Lower birthweight spontaneously conceived individuals and ART that are exposed to Vitrolife culture medium have higher incidence of non-synonymous and rRNA variants. a, b** Prevalence of the haplogroups and sub-haplogroups, respectively, across SC and ART individuals exposed to Vitrolife® culture medium, categorized according to their birthweight percentile. Two-sided Fisher-exact test, no significant differences were found after Bonferroni correction ($p$ values ≤ 0.007 were considered significant). **c** Percentage of SC and ART individuals exposed to Vitrolife® culture medium carrying homoplasmic variants in the different regions and categorized according to their birthweight percentile. Homoplasmies in the tRNA regions were more frequently found in children with a birthweight <P10 (29.4% vs 6.4%, two-sided Fisher's exact test with Bonferroni correction, $p = 0.006$ ($p$ values ≤ 0.00625 were considered significant)). **d** Number of SC and ART individuals exposed to Vitrolife® culture medium, stratified for birthweight percentile under or above P10 or P25, carrying non-synonymous protein-coding or rRNA loci variants. Children born with a birthweight <P25 more frequently carried non-synonymous protein-coding variants (37.5% vs 29.5%, two-

sided Fisher's exact test, *$p = 0.015$). **e** Number of SC and ART individuals exposed to Vitrolife® culture medium, stratified for birthweight percentile under or above P10 or P25, carrying non-synonymous protein coding and/or rRNA loci variants. 52.9% of <P10 and 52.1% of <P25 individuals carry at least one of such variants (vs >P10: 31.3%, two-sided Fisher's exact test, $p = 0.106$, >P25: 28.6%, Fisher's exact test, **$p = 0.002$). **f** Sum of loads of SC and ART individuals exposed to Vitrolife® culture medium under or above P10 and P25 carrying non-synonymous protein coding and rRNA variants. <P10 and <P25 had overall higher cumulative hetero-plasmic loads of non-synonymous and rRNA variants (two-sided Mann–Whitney U test, $p = 0.077$ and **$p = 0.003$, respectively). $n < P10 = 17$ individuals, $n > P10 = 265$ individuals, $n < P25 = 48$ individuals, $n > P25 = 234$ individuals. Horizontal bars in panels represent the mean of the values in the scatter plot, each dot represents a sample. # significant $p$-value after Bonferroni correction, +$p < 0.2$, included in backward conditional binary logistic regression. Source data are provided as a Source data file.

with overall modest changes in heteroplasmic load that were not different between SC and ART individuals (Fig. 4a, Supplementary Table 24), and 109 de novo variants, 36 of which were in non-coding regions, 54 in protein-coding, 8 in rRNA and 11 in tRNA loci (Fig. 4b, Supplementary Data 3). ART mother-child pairs had a higher incidence of de novo non-synonymous variants than SC mother-child pairs (ART: 38.8% (26/67), SC: 20.0% (18/90), Fig. 4c). In line with this, while there were no differences in the sum of heteroplasmic loads of transmitted variants per individual (Fig. 4a), ART children showed higher heteroplasmic loads of de novo non-synonymous variants (Fig. 4d). These findings suggest that the differences in mtDNA variants between ART and SC children are mainly due to a more frequent acquisition of de novo non-synonymous protein-coding variants in ART children.

**Maternal age and the size of the oocyte cohort after OS positively correlate to a higher incidence of de novo mtDNA variants in the children and oocytes**

We hypothesized that the ART-associated procedure of OS may be resulting in the differences in the de novo variants that were observed between ART and SC mother-child pairs. To test this, we compared the mitochondrial genome of sibling oocytes obtained from either natural

menstrual cycles or after OS from 29 young women (Supplementary Table 25). We sequenced a large part of the mtDNA (12.9 Kbp) of 113 individual oocytes, 48 from natural and 65 of OS cycles, along with three sources of somatic DNA from the donors. This allowed us to categorize the heteroplasmic variants as either transmitted (present in at least two samples of the same donor, Supplementary Data 4) or de novo (unique to one oocyte, Supplementary Data 5).

Fourteen transmitted variants were identified in the oocytes of seven donors. No trends were observed in changes in heteroplasmic load of the same variant in oocytes obtained after a natural and an OS cycle (Fig. 5a, Supplementary Data 4 and Supplementary Table 26).

In total, we identified 113 de novo variants, equally distributed over the natural cycle oocytes and OS oocytes (mean 1.7 and 1.8 variants per oocyte for natural and OS oocytes, respectively, Fig. 5b, Supplementary Data 5). There were also no differences in the location of these de novo variants, with similar distributions between natural and OS oocytes in the non-coding regions and in the rRNA, tRNA and protein-coding loci (Fig. 5b). In analogy to the previous analyses, we reduced the complexity of the data by generating the sum of the heteroplasmic loads of the de novo variants in each oocyte (Fig. 5c). We found no significant differences in the heteroplasmic loads of natural and OS cycle oocytes.

**Table 3 | Discriminant analysis to predict a birthweight under the 10th percentile in SC individuals and ART individuals exposed to Vitrolife® culture medium**

| Factors included in model | Standardized Canonical Discriminant Function Coefficients | | Wilks' Lambda N = number of cases included in the model | Correctly classified (original vs cross-validated grouped cases) Correctly classified in each category |
|---|---|---|---|---|
| All pregnancy-related factors | Maternal age | 0.249 | P = 0.010 N = 200 | 89.0% vs 87.0% 90.5% >P10 correct 40% <P10 correct |
| | Smoking | 0.756 | | |
| | Pregnancy hypertension | 0.596 | | |
| | Maternal BMI > 25 | −0.509 | | |
| | Primiparity | 0.132 | | |
| | Gestational diabetes | 0.001 | | |
| mtDNA factors only | Homoplasmic tRNA variants | 0.817 | P < 0.001 N = 282 | 84.0% vs 84.0% 87.2% >P10 correct 35.5% <P10 correct |
| | Heteroplasmic non-synonymous and rRNA variants | 0.764 | | |
| | Haplogroup K1 | 0.302 | | |
| Pregnancy and mtDNA factors | Maternal age | 0.221 | P < 0.001 N = 200 | 87.0%% vs 86.0% 87.9% >P10 correct 70.0% <P10 correct |
| | Smoking | 0.569 | | |
| | Pregnancy hypertension | 0.462 | | |
| | Maternal BMI > 25 | −0.362 | | |
| | Gestational diabetes | 0.043 | | |
| | Primiparity | 0.076 | | |
| | Homoplasmic tRNA variants | 0.700 | | |
| | Heteroplasmic non-synonymous and rRNA variants | 0.179 | | |
| | Haplogroup K1 | 0.233 | | |

**Table 4 | Discriminant analysis to predict a birthweight under the 25th percentile in SC and ART individuals exposed to Vitrolife® culture medium**

| Factors included in model | Standardized Canonical Discriminant Function Coefficients | | Wilks' Lambda N = number of cases included in the model | Correctly classified (original vs cross-validated grouped cases) Correctly classified in each category |
|---|---|---|---|---|
| All pregnancy-related factors | Maternal age | 0.871 | P = 0.174 N = 200 | 62.5% vs 59.5% 63.3% >P25 correct 58.1% <P25 correct |
| | Smoking | 0.412 | | |
| | Pregnancy hypertension | 0.392 | | |
| | Maternal BMI > 25 | −0.109 | | |
| | Primiparity | 0.132 | | |
| | Gestational diabetes | −0.093 | | |
| mtDNA factors only | Homoplasmic tRNA variants | 0.369 | P < 0.001 N = 282 | 64.9% vs 64.9%% 64.5% >P25 correct 66.7% <P25 correct |
| | Heteroplasmic non-synonymous and rRNA variants | 0.681 | | |
| | Haplogroup I | 0.411 | | |
| | Haplogroup J | 0.343 | | |
| | Haplogroup T | −0.398 | | |
| Pregnancy and mtDNA factors | Maternal age | 0.565 | P < 0.046 N = 200 | 68.5%% vs 63.0% 68.6% >P25 correct 67.7% <P25 correct |
| | Smoking | 0.326 | | |
| | Pregnancy hypertension | 0.217 | | |
| | Maternal BMI > 25 | −0.102 | | |
| | Gestational diabetes | −0.022 | | |
| | Primiparity | 0.115 | | |
| | Homoplasmic tRNA variants | 0.333 | | |
| | Haplogroup I | 0.218 | | |
| | Haplogroup J | 0.236 | | |
| | Haplogroup T | −0.325 | | |
| | Heteroplasmic non-synonymous and rRNA variants | 0.531 | | |

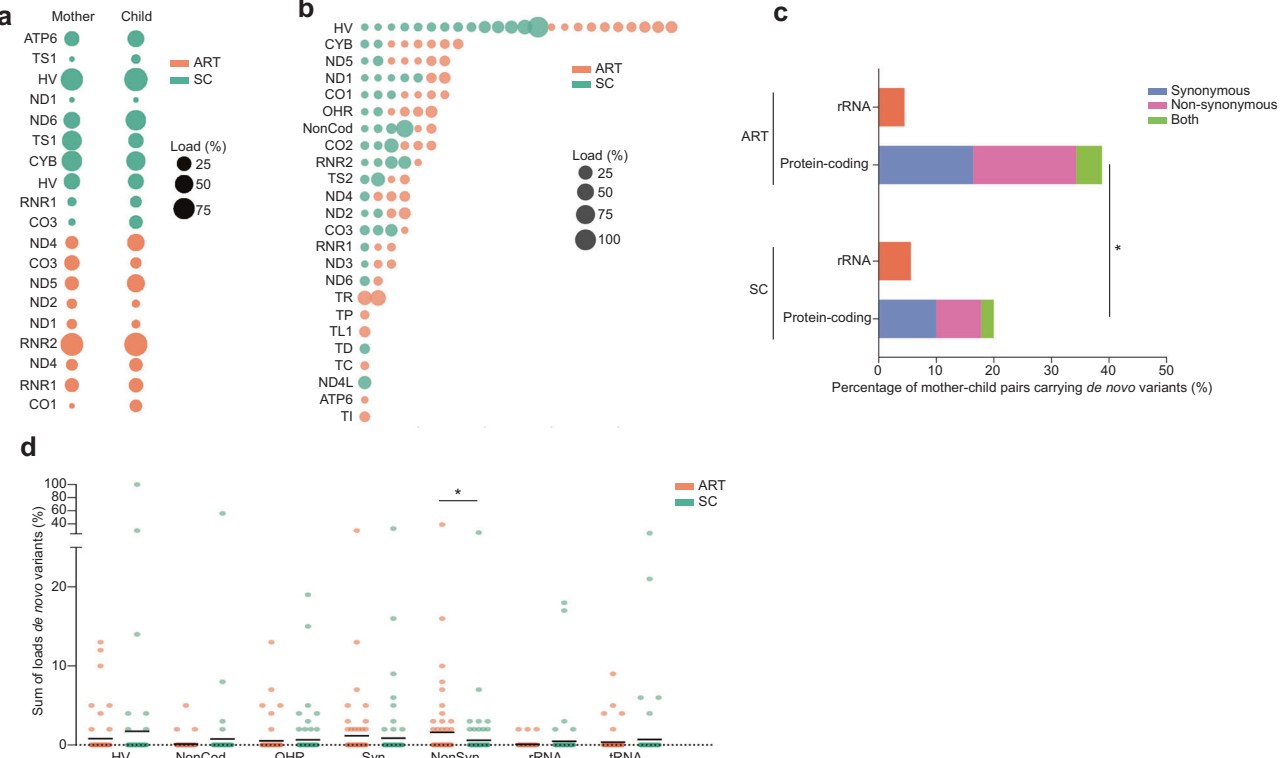

**Fig. 4 | ART individuals have a higher incidence of de novo non-synonymous mtDNA variants than their spontaneously conceived peers. a** Overview of the transmitted heteroplasmic variants identified in the mother-child pairs, categorized according to mode of conception and labelled according to their location in the mtDNA genome. The size of the bubble is proportional to the heteroplasmic load in the mother and her child. No statistically significant differences between SC and ART, two-sided Mann–Whitney U test. **b** Overview of de novo heteroplasmic variants identified in the mother-child pairs, categorized according to mode of conception and labelled according to their location in the mtDNA genome. The size of the bubble is proportional to the heteroplasmic load in the child. **c** Number of ART and SC mother-child pairs showing de novo non-synonymous and rRNA variants. ART children often show more de novo protein coding variants than SC children (two-sided Fisher's exact test, *p = 0.01). **d** Sum of the heteroplasmic loads of the de novo variants in ART and SC individuals, categorized per region. Two-sided Mann–Whitney U test *p = 0.048, n ART = 67 individuals, n SC = 90 individuals. Horizontal bars represent the mean of the values in the scatter plot, each dot represents a sample. Source data are provided as a Source data file.

Finally, we used a regression utilizing the Poisson Generalized Linear Model to test if the number of de novo variants in mother-child pairs and in the oocytes correlated to other factors, such as the maternal age at the time of conception, the FSH units used during the OS and the size of the cohort of the retrieved oocytes. For the oocytes, we used the average number of de novo variants per oocyte for each donor. The FSH units did not correlate to the de novo variants in any of the models we tested, and were further excluded from the analysis. In the cohort including the oocytes and the mother-child pairs (N = 170), we found that maternal ageing increased the total number of de novo variants (Fig. 5d and Table 5) and of rRNA variants (Table 5) but not of synonymous or non-synonymous variants (Supplementary Table 27). The models tested on the oocytes (N = 113) showed that both maternal age and the number of oocytes retrieved after OS positively correlate to the total number of de novo mtDNA variants in the oocyte (Fig. 5e, f, Table 5 and Supplementary Table 27). While the maternal age associates most strongly with the total number of de novo mtDNA variants and to rRNA variants, the number of oocytes retrieved positively correlates to the number of de novo non-synonymous and rRNA variants in an oocyte, suggesting two different mechanisms of action. Interestingly, the maternal age was significantly higher in the ART group (Table 1). Hence, the combination of undergoing OS with having an older maternal age could explain why ART children have a higher incidence of carrying de novo non-synonymous and rRNA variants than their SC peers.

## Discussion

This study identifies, for the first time, genetic differences between ART and SC children that are associated with their birthweight. We found that the mtDNA of ART children more frequently presents variants in protein-coding and rRNA regions, and that they more often carry de novo non-synonymous variants than their spontaneously conceived peers. Using regression and discriminant analysis models, we found that these non-synonymous and rRNA variants associate with a higher chance of being born under the 25th birthweight percentile in both ART and SC groups, but with a stronger association in the latter. In ART children, the culture medium to which they were exposed during in vitro embryo culture before transfer was found to be an important confounding factor, especially in older now obsolete culture media. Finally, we found that maternal ageing and, for ART individuals, the size of the oocyte cohort retrieved during OS, were the primary factors associated with the incidence of de novo variants in the children and oocytes.

The exact mechanisms for the association between mtDNA variants and birthweight are to be elucidated, but the impact of mtDNA variation on health and disease is well-documented[62,74,75]. Inherited disease-causing mtDNA variants are linked to low birthweight, obesity and insulin resistance[40–42], and mitochondrial haplogroups have been associated with a lower bioenergetic fitness[76], subfertility[77] and obesity[45,50]. A number of homoplasmic and heteroplasmic variants have been reported to be associated with body composition, although we were not able to replicate this finding in our study[43–49]. Conversely,

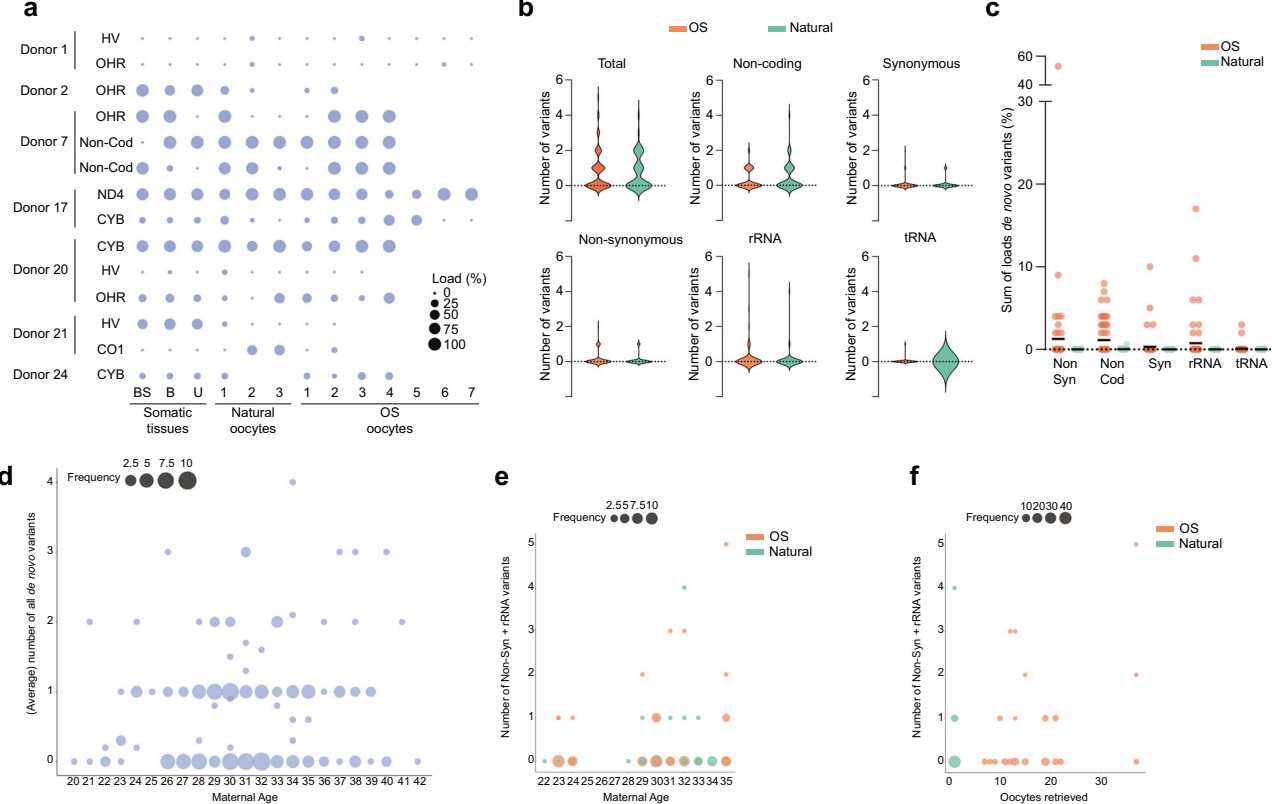

**Fig. 5 | Maternal ageing and the size of the oocyte cohort after OS associate with a higher incidence of de novo mtDNA variants in the children and oocytes.** **a** Transmitted heteroplasmic variants identified in the oocyte donors and their oocytes. The size of the bubble is proportional to the heteroplasmic load of the variant in each of the samples, which include somatic tissues and up to 3 oocytes in a natural cycle and up to 7 oocytes after OS. **b** Distribution of the number of oocytes after natural and OS cycles showing de novo variants in the different regions. No statistically significant differences were observed (two-sided Chi-square tests). **c** Sum of the heteroplasmic loads of the de novo variants in oocytes after natural and OS cycles, categorized per region. *n* OS = 65 oocytes, *n* Natural = 48 oocytes. Horizontal bars in panel represent the mean of the values in the scatter plot, each dot represents a sample. **d** Total number of de novo variants in mother-child pairs and in oocytes plotted against the maternal age at time of conception or oocyte collection. Poisson Generalized linear model regression (PGLMR) shows that maternal ageing correlates significantly with the total number of de novo variants in the next generation (*p* < 0.001). **e** Number of de novo non-synonymous and rRNA variants in oocytes, plotted against the maternal age at time of oocyte collection. PGLMR shows that maternal ageing does not significantly correlate with the number of de novo non-synonymous and rRNA variants in oocytes (*p* = 0.07). **f** Number of de novo non-synonymous and rRNA variants in oocytes, plotted against the number of oocytes retrieved in a cycle. PGLMR shows that the size of oocyte cohort significantly correlates with the de novo number of non-synonymous and rRNA variants in oocytes (*p* = 0.007). BS: buccal swab, B: blood, U: urinary tract cells, In (**d**–**f**), the size of the dots is proportional to the number of samples with the same data point. Source data are provided as a Source data file.

the obesity-associated haplogroup T[45,50], was under-represented in the lower birthweight individuals, and although the statistical significance was lost after correcting for multiple testing, these results are

**Table 5 | Generalized linear model regressions with Poisson distribution on de novo variants in mother-child pairs and oocytes**

| | (B) | 95% Wald C.I. | Significance |
|---|---|---|---|
| **Total number of de novo variants in mother-child pairs and oocytes (*N* = 170)** | | | |
| Maternal age | 0.037 | 0.024–0.050 | <0.001 |
| **rRNA de novo variants in mother-child pairs and oocytes (*N* = 170)** | | | |
| Maternal age | 0.054 | 0.10–0.097 | 0.015 |
| **Total number of de novo variants in oocytes (*N* = 113)** | | | |
| Maternal age | 0.079 | 0.031–0.127 | 0.001 |
| Oocytes retrieved | 0.017 | −0.004–0.037 | 0.110 |
| **Total number of de novo non-synonymous and rRNA variants in oocytes (*N* = 113)** | | | |
| Maternal age | 0.033 | −0.003–0.068 | 0.074 |
| Oocytes retrieved | 0.021 | 0.006–0.036 | 0.007 |

supporting the role of haplogroup T in modulating body weight. In our study, we find an association of birthweight with the presence of non-synonymous and rRNA heteroplasmic variants with a relatively low load and of unknown pathogenic potential, as they have not been associated with disease. Healthy individuals are known to carry heteroplasmic variants, most of which are not considered to be disease-causing[78]. Nevertheless, these heteroplasmic variants can be a source of differences in health status and progression of ageing[75]. For instance, variants inducing an amino-acid change or alterations in the transcription machinery can decrease the efficiency of the oxidative phosphorylation and increase the production of reactive oxygen species[79]. Heteroplasmic variants are well-documented to play a key role in ageing[80–82], neurodegenerative diseases and cancer[83], and combinations of different non-pathogenic variants at low heteroplasmic loads, such as found in this study, have been proven to have a synergistic negative effect on mitochondrial function[84].

The study of the cohort of mother-child pairs showed that the differences between ART and SC individuals were mainly due to increased maternally transmitted protein-coding and rRNA variants, in combination with more de novo non-synonymous protein-coding variants in the ART individuals. We hypothesized that OS could be responsible for this increase in de novo variants, but in first instance we

found no differences when directly comparing the mtDNA variants in natural and OS cycle oocytes. Conversely, we found that maternal ageing positively correlated with the number of de novo variants found in the next generation and that the size of the cohort of oocytes retrieved after OS positively correlated to the number of non-synonymous and rRNA variants in the oocyte. The relationship we find between maternal age and the number of de novo mtDNA variants in offspring and oocytes is fully in line with other studies carried out on mother-child pairs[85,86] and on mouse and macaque oocytes[87,88], where a clear association between maternal age and mtDNA mutational load has been identified. Given the significant effect of maternal age on birthweight that we and others have identified[89–91], it seems plausible that an increase in de novo mtDNA variants in the oocytes may be one of the mechanisms by which advanced maternal age results in lower birthweight children.

In ART children, next to maternal ageing, the size of the oocyte cohort associates with a higher incidence of non-synonymous and rRNA variants, a link that, to our knowledge, has not been previously reported. A potential explanation for this observation is that the procedure of OS abrogates the competition for FSH between the different follicle-oocyte complexes initiating maturation in a menstrual cycle. This lack of competition may allow for the ovulation of oocytes carrying detrimental mtDNA variants, such as non-synonymous changes and in rRNA loci, that are otherwise outcompeted by the fitter dominant oocyte in a natural menstrual cycle[86,92,93]. A few studies have investigated the association between OS and birthweight, with contradictory results. Natural cycle ART, without OS, has been found to associate with higher birthweights than cycles with OS by Mak et al.[94], but not by Sunkara et al.[95]. Similarly, the number of oocytes retrieved has been found to associate with lower birthweights in some cohorts[96] but not in others[97]. The reasons for these discrepancies remain to be elucidated and may be, at least in part, due to the significant impact of culture media on embryonic development and birthweight[25–28,34,70,98].

Lastly, it is important to bear in mind the limitations of this study. First, we report on two cohorts of samples, recruited in two different centres, that, when analysed separately, do not achieve statistical significance for the findings we report. Second, in this study cohort, there are no differences in birthweight between ART and SC individuals. It is likely that the differences in mtDNA genotype between ART and SC individuals (Fig. 1) would have been more pronounced if the birthweight in these two groups would have been significantly different. A potential factor influencing this aspect is that the ART individuals were exposed to three different culture media, which clearly have an important and differential impact on birthweight. Third, the associations we find between haplogroups and birthweight are likely underpowered and should be taken with caution, as they do not retain statistical significance in all models we tested. Moreover, while the model to discriminate the 10th birthweight percentile is robust, the model for the 25th birthweight percentile is only just statistically significant, and will require further adjustment and improvement. Fourth, the variants we report to associate with birthweight have no proven pathogenic effect beyond what in silico models can provide, and further research will be needed to demonstrate a molecular link. Finally, it is important to consider that when analysing only one tissue per individual, it cannot be excluded that the identified heteroplasmic variants are tissue-specific, in which case they had a limited potential to affect the individual's birthweight. Taken together, future replication studies on larger cohorts will be key to validating our findings and will allow for the refinement of the birthweight prediction models.

In sum, our study has identified a link between the presence of non-synonymous mtDNA heteroplasmic variants in protein-coding regions and rRNA loci and lower birthweight, which are more common in children from older mothers and that are born after an ART treatment. We propose that these variants may result in a modest but still sufficient mitochondrial dysfunction to result in a lower birthweight

percentile, providing the first evidence for mitochondrial genetic factors that could explain the differences observed between ART and SC individuals. The long-term health consequences of these changes remain to be studied to establish how these findings will impact clinical practice and patient counselling in the future.

## Methods

For all parts of this study, the design and conduct complied with all relevant regulations regarding the use of human study participants, and all were approved by the local ethical committee of the UZ Brussel and the ethical review board of the Maastricht University Medical Centre. All individuals provided written and signed informed consent. The participants consented to the publication of clinical parameters in an anonymized manner. The study was conducted in accordance to the criteria set by the Declaration of Helsinki.

### Recruitment and donor material

The participants were recruited at the Universitair Ziekenhuis Brussel (UZ Brussel, Belgium) and at the Maastricht University Medical Centre (MUMC, Netherlands). All participants provided informed consent and did not receive a compensation unless stated differently. All ART children were born from fresh embryo transfers at the cleavage stage, details on the use of intracytoplasmic sperm injection (ICSI) or in vitro fertilization is known for only 150 of the 270 ART cycles. This information was not included in the analysis because ICSI has been found not to increase the risk for adverse perinatal outcomes[99]. The DNA samples extracted from blood from newborns were supernumerary material recruited at the UZ Brussel for a study designed to test if paternal mitochondrial DNA can be found in individuals born after ISCI[100]. The parents gave informed consent. The ethical commission gave permission to utilize the archived DNA samples for this study, but not the associated clinical parameters. The 9-year-old samples were recruited at the MUMC, as part of a follow-up study on a prospective observational cohort study[25]. The parents were approached after the ninth birthday of their child to participate in this study. The local ethics committee approved the study and both parents of all children gave written informed consent. The DNA was extracted from saliva samples. The placental DNA samples were collected for a study coordinated by the MUMC[26,34,70]. Ethical approval was requested but was waived, as in accordance with Dutch law, spare placenta tissues can be used for research after informed consent of the patient, without further permission of an ethical committee since no interventions were needed to obtain the samples. The placental biopsies were obtained in six IVF clinics in the Netherlands (Amsterdam UMC, location AMC in Amsterdam, Catharina Hospital in Eindhoven, St. Elisabeth Hospital in Tilburg, Maastricht University Medical Center in Maastricht and University Medical Center Groningen in Groningen) and five IVF clinics affiliated to these hospitals, all patients signed informed consent[26]. Details on sample collection and extraction can be found in previously published work[26,34]. The 18-year-old ART individuals, the samples were collected as part of a larger project on the cardio-metabolic and reproductive health of young adult women and men born after ICSI conducted at the UZ Brussel[101,102]. These individuals are part of a cohort that has been prospectively monitored since birth. All parents of eligible ICSI offspring in our database were sent a letter explaining the background and set-up of the study. Shortly after, these parents were contacted by phone in order to explore their and their children's willingness to participate. After parental consent was obtained, the young adults were approached directly and invited to participate. All participants signed informed consent and received an incentive by means of a gift voucher. The DNA was extracted form peripheral blood. The local ethical committee of the UZ Brussel approved this study. The 18-year-old SC individuals were recruited first-year students at the campus of the Vrije Universiteit Brussel. They were approached on campus, where they received an information brochure on the study, an

informed consent form and the necessary materials to provide a buccal swab. The volunteers that enrolled in the study signed the informed consent prior to providing the sample. This study was approved by the local ethical committee of the UZ Brussel.

For the study of oocytes, twenty-nine young women donated oocytes in up to three natural menstrual cycles and after one OS cycle. All donors were recruited at the Center for Reproductive Medicine of the UZ Brussel, as part as the standard oocyte donation programme, where also all medical procedures took place. All donors provided informed consent for all medical procedures and for the use of their oocytes for research, and received financial compensation. The local ethics committee of the UZ Brussel approved all parts of this study. The inclusion criteria for the recruitment were: 19–35 years old, healthy and with a body mass index of 18–32. The exclusion criteria were: any type of medical or genetic condition that may interfere with the health of the oocytes, American Fertility Society endometriosis grades (AFS) III-IV, ovulatory disorders belonging to World Health Organisation (WHO) categories 1–6, body mass index of <18 or >32, use of medication that may interfere with the oocyte competence and psychiatric disorders. The donors underwent two types of donation cycles: up to three natural cycles and one cycle after OS (Supplementary Table 25). During the natural cycle oocyte donation, a blood sample was taken on day 2 of the follicular phase to assess several parameters: serum estradiol, progesterone, luteinizing hormone, follicle-stimulating hormone (FSH), and human chorionic gonadotrophin (hCG) levels. On day 10, a blood sample was taken again to check the same parameters except hCG as well as a vaginal ultrasound to assess follicular development. If not sufficient, additional blood samples and ultrasound were taken. The oocyte retrieval was performed 32 h after administration of 5000 IU exogenous hCG. The OS cycle started on the first day of menstruation following the natural cycle. Starting on day 2 of the cycle, a gonadotrophin-releasing hormone (GnRH) antagonist protocol was applied, using recombinant FSH if hormonal analysis was within normal limits on day 2. The starting dose of the recombinant FSH varied between 150–300 IU per day. At day 6 of stimulation, a GnRH antagonist (Ganirelix; Orgalutran®; MSD, Oss, The Netherlands) with a dose of 0.25 mg per day was given. Final oocyte maturation was achieved by administration of a GnRH agonist (Gonapeptyol®, Ferring Pharmaceuticals, St-Prex, Switzerland) with a dose of 0.2 ml when at least 3 follicles reached a mean diameter of 17–20 mm. Oocyte retrieval was performed according to routine procedures, 36 h after trigger.

## DNA isolation and long-range PCR protocol for mtDNA enrichment

DNA was extracted from peripheral blood, placenta, saliva and buccal swab samples, and isolated from single oocytes as previously described by our group[103,104]. The mtDNA was enriched by long-range PCR using two primer sets to cover the full mtDNA in bulk DNA samples. For single oocytes, only the first primer set was used. The sequences of the first primer set (5042f-1424r) were 5′-AGC AGT TCT ACC GTA CAA CC-3′ (forward) and 5′-ATC CAC CTT CGA CCC TTA AG-3′ (reverse) generating amplicons of 12.9 Kb. The sequences of the second primer set (528f-5789r) were 5′-TGC TAA CCC CAT ACC CCG AAC C-3′ (forward) and 5′-AAG AAG CAG CTT CAA ACC TGC C-3′ (reverse) generating amplicons of 5.3 Kb. Both primer sets were purchased from Integrated DNA Technologies, were diluted to 10 μmol and kept at −20 °C. The master mix for the long-range PCR was prepared as follows: 10 μL of 5x LongAmp buffer (LongAmp Taq DNA Polymerase kit M0323L, New England Biolabs, stored at −20 °C), 7.5 μL dNTPs (dNTP set, IllustraTM, diluted to 2 mM and stored at −20 °C), 2 μL of the forward and reverse primers (10 μM), 2 μL of Taq Polymerase LongAmp (5 units) and 50 ng of diluted DNA (working solution: 10 ng/μL) in a total of 50 μL. In case of the oocytes, a single-cell long-range PCR was applied. Here, only the first primer set was used and 2.5 μL of Tricine

(Sigma-Aldrich T9784, diluted to 200 nM and stored at 4 °C) was added to buffer the alkaline lysis buffer (200 mM NaOH and 50 mM DTT) used in the oocyte collection. The master mix was directly added to the sample. Successful PCR amplification was confirmed by running a gel-electrophoresis. Specificity of the primer sets for the mitochondrial genome was controlled by amplifying DNA of RhoZero cells, which do not contain mitochondrial DNA. More detailed protocols of the long-range PCR and the extensive validation of the method were published previously[104,105].

## Preparation of samples for sequencing and data analysis

After PCR amplification, the samples were purified with AMPure beads and the amplicons from both primer sets were pooled together in a 0.35/0.65 ratio (35% of the amplicon from the second primer set and 65% of the amplicon of the first primer set). Next, the library was prepared with the KAPA HyperPlus kit and purified with AMPure beads. The quality of the library was checked by electrophoresis on the AATI Fragment Analyzer using the HS NGS Fragment kit. Next, the libraries were diluted to 2 nM in EBT buffer and pooled before sequencing on the Illumina platform. The raw sequencing data is considered personal data by the General Data Protection Regulation of the European Union (Regulation (EU) 2016/679) and cannot be publicly shared. The data can be obtained from the corresponding author upon reasonable request and after signing a Data Use Agreement. Alignment to the reference genome (NC_0.12920.1) was done using BWA-MEM. The generated bam files were uploaded to the online platform "mtDNA server" to determine the haplogroup and to identify other homoplasmic and heteroplasmic variants[106] (https://mitoverse.i-med.ac.at/) and MuTect2[107] was used to detect small insertions and deletions. Variants with ≥1.5% heteroplasmic load (i.e. more than or equal to 1.5% of total sequenced molecules carried the variant), or ≥2% in case of the single oocytes, were annotated and possible amino-acid changes were identified using MitImpact2[108]. Homoplasmies were defined as variants with a load >98.5% (100% minus the detection limit) for the samples of the children and the mothers, for the oocytes this was >98%. We excluded variants that were called in a set of different regions that were prone to generate PCR and sequencing errors, resulting in 1037 bp being excluded from the final analysis (Fig. 1a, Supplementary Table 28). The variants identified in all the samples included in this study are listed in the supplementary data file, including the predicted pathogenicity scores of variants that are non-synonymous protein-coding or tRNA-coding. The pathogenicity scores were obtained through in silico analysis using MutPred2[71] for the non-synonymous variants and MitoTip[109] for the tRNA variants. MutPred2 scores >0.61 and MitoTip scores >50% were considered as potentially pathogenic. This method, including PCR, sequencing and data analysis, was previously set up in our lab and thoroughly validated with a lower detection threshold (heteroplasmic level) of 2% for single cells and 1.5% for bulk material, based on multiple analysis of the same samples, including numerous single cells and mixes of DNA samples to mimic multiple levels of heteroplasmy[105,110,111]. All mitochondrial DNA variants called in all samples are provided with this paper in the Supplementary Data 6 file.

## Statistics

Statistical analysis was carried out using SPSS (IBM). The heteroplasmic variants were analysed through an orthogonally rotated exploratory factor analysis, which extracts the variability among variables in the form of independent latent variables (more details can be found in the supplementary information file, in the Supplementary Fig. 1, Supplementary Table 29 and Supplementary Fig. 2). Binary logistic regression and discriminant analysis were used to build models to predict the effect of different variables on the birthweight percentile. For these, samples with missing data were excluded from the analyses. Other statistical analyses were done using the Fisher's exact,

Mann–Whitney U and Student t-test. All statistical tests were two-sided and p-values < 0.05 were considered significant. Correcting for multiple testing was done using the Bonferroni method, and the adjusted threshold for significance is indicated where relevant.

## Reporting summary

Further information on research design is available in the Nature Portfolio Reporting Summary linked to this article.

## Data availability

The raw sequencing data is considered personal data by the General Data Protection Regulation of the European Union (Regulation (EU) 2016/679) and cannot be publicly shared. The data can be obtained from the corresponding author upon request and after signing a Data Use Agreement. All mitochondrial DNA variants called in all samples are provided with this paper in the Supplementary Data 6 file, as well as the clinical parameters associated to the samples. Source data are provided with this paper.

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

## Acknowledgements

The authors would like to express their sincere gratitude towards all staff of the Center for Reproductive Medicine in the UZ Brussel for their engagement in this study, and to thank Hatice Satilmis, Jo Bossuyt and Khadija Rhioui for their support in data collection for this paper. We also thank Iain Johnston and Joanna Poulton for the critical reading and valuable scientific contributions to our work. This research was funded by the Research Foundation Flanders (FWO, Grant numbers 1506617N and 1506717N, to C.S.), by the Wetenschappelijk Fonds Willy Gepts of the UZ Brussel (Grant numbers WFWG14-15, WFWG16-43, WFWG19-19, to C.S.), the Methusalem Grant of the Vrije Universiteit Brussel (to K.S.), and by March of Dimes (6-FY13-153 to A.P.A.V.M). M.R. and E.C.D.D. are predoctoral fellows supported by the FWO, Grant Numbers 1133622N and 1S73521N, respectively.

## Author contributions

J.M., F.B., A.P.A.V.M., M.B., K.St. and F.Z. recruited the study participants, prepared samples and collected clinical parameters. H.V.D.V.,

H.T. and C.B. recruited the oocyte donors and were responsible for the medical procedures for oocyte retrieval and collection. J.M. and F.Z. carried out sample preparation for DNA sequencing. J.M. and E.C.D.D. carried out the bioinformatic analysis of the sequencing data. J.M. consolidated and curated the dataset. M.R. created all figures in the manuscript and contributed to the statistical analysis. K.B. advised on all statistical methods and devised the factor analysis for the mitochondrial variants. S.S. and H.S. contributed to mitochondrial variant interpretation and overall study design. K.Se. provided funding for this study and contributed to overall study design. J.M. and C.S. carried out statistical analysis, interpreted the results and co-wrote the manuscript. C.S. designed the study, provided funding and supervised the study. All authors proofread the manuscript and approved its final version.

## Competing interests

The authors declare no competing interests.
