## [Peer Review File · Nature Communications]

Children born after assisted reproduction more commonly carry a mitochondrial genotype associating with low birthweightREVIEWER COMMENTS

Reviewer #1 (Remarks to the Author):

This article is exploring the effect of heteroplasmic mitochondrial variants load on the birth weight of children born using assisted reproductive technology (ART). The gap in knowledge the authors are attempting to close is important to address given the increase in use of ART. Overall the article is well structured but lacks clarity and there are methodological issues.

Major points

1. It is unclear from the article what the authors aim to test. Are they testing whether heteroplasmic load is associated with birth weight, adjusting for confounders, or are they trying to build a prediction model that could be used for diagnostic purposes. There is also little explanation on why switch to discriminant analyses other than negative results from the logistic model. For example, it would have been helpful to show power calculations in order to judge whether the logistic regression would have been able to detect an effect in the first place.
2. Could the authors give a more detailed description of the discriminant analyses they did. For example, how they decided on $p < 0.2$ for inclusion of variables in the model and how was the cross-validation done. Also, although categorical variables can be used in discriminant analyses, this is not ideal as it violates the normality condition and could cause instability. Further to this, it would help to show what is the added value, if any, of the genetic variants. It would help if the authors report results of discriminant analyses that do not include the genetic variants.
3. Given that quantitative traits afford better power, it may be worth as an initial model to use birth weight as a quantitative trait rather than dividing the weight in categories.
4. The authors should clearly define what they mean by pathogenic (e.g. causing known mitochondrial conditions versus variant increasing the risk of being born with lower birth weight), especially since they are looking at a complex trait.
5. In relationship to the above and since common mitochondrial variants have been associated with BMI and body composition parameters in adults, the authors should specify whether there was any difference between ART and SC children in the distribution of such previously associated variants.
6. It would be helpful to add a paragraph discussing the limitations to the study in the discussion section.

Minor points

1. Is it possible for the authors to call mitochondrial copy number from the data they have. It would be interesting to see if there are any differences between ART and SC children and whether copy number affects birth weight.

2. It would be helpful if the authors used a clearer way to represent the results in figure 1d. The figure is difficult to read.
3. Line 353 change “are predictive for” to “associated with”
4. In the section exploratory factor analyses within supplementary methods, could the author add a figure showing how the eight variables used in the analyses were distributed and also how they correlate with each other. That would help to judge the normality and linearity of the data.
5. Could the authors please re-format the supplementary data tables. They are very difficult to follow.

Reviewer #2 (Remarks to the Author):

In this manuscript, Mertens et al analyze mtDNA sequencing data from 270 individuals born via assisted reproductive technologies (ART) and 181 from spontaneous conception (SC), as well as from 157 mother-child pairs and 113 oocytes from cycles with or without ovarian stimulation (OS). For each individual, they compute the sum of heteroplasmy level for eight variant categories, and use exploratory factor analysis to identify four factors that represent the majority of variance across the eight categories. Four factor scores are then calculated for each sample, using the sum of the heteroplasmy level across categories, which are used for majority of the analyses and significant conclusions.

The authors identify significant differences in mtDNA variation between groups. They identified higher factor 2 scores in ART than SC individuals, and in SC individuals who were $P < 10$ or $P < 25$ birth weight. They also note SC $P < 25$ individuals more frequently carried variants in proteins and rRNAs, and that SC and female ART individuals $P < 25$ showed a higher cumulative heteroplasmic load of these variants. They also reported significantly higher factor 2 scores in ART individuals for transmitted but not de novo variants. They conclude that individuals born after ART more frequently carried protein and rRNA variants with higher loads, and that these differences were predictive of birth weight.

Overall, the hypothesis is interesting, and a link between mtDNA variation and birth weight in ART and SC individuals would be noteworthy and significant. However, many of the major results of this study require additional evidence to support the conclusions made.

Major:

- Several conclusions are based solely on the factor scores, which have no validated utility for assessing the impact of mtDNA variants. Rather, these factor scores apply somewhat arbitrary weights to variant categories. Many conclusions using factor 2 scores rely on the inclusion of synonymous (i.e. under ‘protein variants’). Synonymous variants do not change protein sequence and can be regarded as

evidence of benign impact per clinical variant classification guidelines (e.g. PMID: 32906214). Significance tests which compare different frequencies or heteroplasmy levels for protein and rRNA variants between groups should be repeated with synonymous excluded (i.e. limit to non-synonymous and rRNA). This would help strengthen the claim that rRNA and protein variants are driving the association with birth weight through impacts on mitochondrial function. Can any other evidence be provided, such as population frequency data, to strengthen the conclusions that protein and rRNA variants are driving low birth weight through functional effects? Also, why are tRNA variants systematically excluded from most analyses?

- The authors state their motivation was to investigate the impact of mtDNA variation on the increased risk of lower birth weight in ART vs SC, however their cohort shows no significant difference in P<10 and P<25 categories between the ART and SC groups, per Table 1. How does this impact their major conclusion that mtDNA variants drive low birth weight and that this is more common in ART individuals?

- Lines 233-236: “This indicates that individuals born after ART tend to carry higher loads of heteroplasmic variants in the protein-coding and/or rRNA regions” –the authors should provide support for this conclusion drawn from the factor scores by showing the distribution of the heteroplasmy level of variants in each category, and statistical assessment of their difference between ART and SC. In Table S6 the proportion of heteroplasmic variants in rRNA genes is very similar between ART and SC (8.7% vs 8.0%), implying that any impact must be in their heteroplasmy level. However most rRNA variants detected appear to be <20% heteroplasmy level per Figure 1d, a mutant load at which functional impacts are particularly unclear. The proportion of ART and SC individuals with heteroplasmies, across variant categories, should also be shown as done for homoplasmies in Figure 1b and Table S5.

- Table 1 lists a range of tissue types analyzed. Did the ART and SC individuals have multiple tissues analyzed? Was the distribution of sample types used for DNA similar between ART and SC? Given sample type can influence heteroplasmy, it would be important to ensure results are not driven by sample type differences between groups.

- Lines 245-249: “Conversely, SC individuals who were <P10 or <P25 had significantly higher scores in factor 2 of the heteroplasmic variants analysis than children born at >P10 or >P25... In line with this, SC <P25 individuals more frequently carried variants in protein and rRNA loci” – the authors should provide data on the proportion of SC individuals carrying heteroplasmies for each variant categories to support this, as Table S11 referred to does not appear to show this. For Table S11 it would be helpful to state in the legend exactly what the % and () represent and how those percentage values were calculated.

- Line 259-263: “...male individuals that were subjected to Cook and UZB medium (e.i. an in-house medium made in the UZ Brussel) are more frequently born with a birth weight <P25 than females (25.5% in males vs 10.7% in females). Remarkably, only 7.7% of these <P25 males carried variants in protein-

and rRNA-coding loci as compared to 30.0% of the <P25 females” – the authors should provide the data underlying this observation (i.e. supplementary table), as it forms the basis of excluding male ART individuals from all subsequent analyses. Statistical analysis on the difference in birth weights between males and females across all culture mediums used would also be of interest to support the effect of the media on males and their exclusion.

- The discriminant analysis showing mtDNA variants in combination with other data could predict birth weight category is a particularly interesting result. Does the classification precision decrease when mtDNA variants are removed from the model, i.e. how much of this is related to mtDNA vs factors such as maternal age, hypertension and or smoking? This is important for their assertion that “differences in the mitochondrial genome were also predictive of the risk of a lower birth weight percentile”. For assessment of confounding factors/parameters for the logistic model, listed in Table S12, it appears that assessment of homoplasmies was restricted to tRNA and heteroplasmies to protein plus rRNA. The authors should provide the values for all categories of homoplasmies and heteroplasmies (i.e. per Table S11) or provide a sound justification for their exclusion from the logistic regression models.

- Lines 292-303: “We found a higher incidence of ART mother-child pairs presenting de novo variants in the protein-coding regions....The scores for factor 2 (driven by the presence of protein-coding and rRNA variants) were significantly higher in ART individuals for the transmitted but not for the de novo variants ... These findings suggest that the differences in mtDNA variants between ART and SC children are due to a higher transmission of protein- and rRNA-coding variants” - The authors should provide support for this conclusion drawn from the factor scores by showing data on the transmission and de novo origin of variants across each category for ART and SC individuals (i.e. proportion of individuals with said variants and differences in their heteroplasmy level, especially for transmitted where no data is presented), and statistical assessment of any difference between groups. How do they contextualize the higher incidence of protein de novo in ART with the factor 2 scores only being significant for transmitted but not de novo?

- Lines 317-319: “No trends were observed in changes in heteroplasmic load of the same variant in oocytes obtained after a natural and an OS cycle (Figure S3 in supplementary appendix).” – it appears that heteroplasmic load was more likely to decrease in oocytes from natural vs OS cycle (48.5% vs 31% in Figure S3). Can the authors explain why they regarded this data to represent no change?

- Lines 331-333: “the location of the de novo variants as well as the number of oocytes carrying de novo protein-coding and rRNA variants, was similar between the natural and OS cycle oocytes (Figure 3d)” – Figure 3d shows approximately two-fold increased rRNA and non-synonymous de novo variants in OS vs natural, was statistical testing used to conclude they were similar?

- Lines 347-350: “the maternal age was significantly higher in the ART group (table I, t-test, $p < 0.0001$), potentially explaining why ART children have a higher incidence of de novo variants” - The authors do not

present data showing that there is a higher overall de novo rate in ART vs SC individuals – this should be provided to support their assertion.

- Lines 354-355: “the presence of protein-coding and rRNA mtDNA variants was predictive for the birth weight percentile of the females but not the males” – can the authors point to their data showing the assessment of variant predictive value in males vs female birth weight? The discriminant analysis used to classify individuals (Figure 2j-k) appeared to only be performed on females.

Minor:

- The method used to determine haplogroups, and haplogroup variants, should be stated.
- More than 1kb of the total mtDNA is excluded from analysis, this should be reiterated in the results text and highlighted in Figure 1d.
- Do the authors have a rationale for why they segment intergenic variants into four groups (HV, non-coding, OHR and TAS)? This would appear to reduce power for analyses across intergenic variants.
- Line 215 “individuals frequently carried multiple variants with different heteroplasmic loads” appears at odds with the average of ~1 heteroplasmic variant per individual reported. Can the authors clarify?
- It should be remarked whether the statistical tests applied were one or two sided, as appropriate.
- In general the factor score dot plots are hard to compare, could median or mean bars be added or the jitter increased (to avoid overlap of points). What does the horizontal line represent?
- Lines 322-324: “To determine whether the de novo variants identified in multiple oocytes of the same donor could be considered as independent events in the statistical analysis” – do the authors mean different de novo mutations? As if the same de novo variant, then wouldn’t identification in multiple oocytes from the same donor be consistent with it not being a true de novo but rather under the limit of heteroplasmy detection?
- Lines 374-375: “The exact mechanisms for the association between non-disease causing mtDNA variants and birth weight are yet unclear” – the authors should clarify in the results text if disease-causing variants were excluded, and how these were defined.

Reviewer #3 (Remarks to the Author):

The manuscript by Mertens and her colleagues addresses a very important topic - the health risks of children born following the use of assisted reproduction techniques (ART). The approach of this work is very interesting and complex, using DNA samples from ART and spontaneously conceived (SC) individuals, mother-child pairs, and the study of individual oocytes. Whereas the oocytes were obtained

both as a result of a natural and a stimulated cycle. The obtained DNA samples were analyzed for inherited mutations in mitochondrial DNA and for inherited and de novo heteroplasmy.

The results of the study are intriguing, i.e., ART children more frequently carry variants with higher heteroplasmic loads. Heteroplasmy in ART children are either maternally inherited or represent de novo mutations, primarily in mitochondrial genes encoding proteins and rRNA. In addition, heteroplasmy is more common in both ART and SC of small for gestational age children. Moreover, the single oocyte analysis revealed that heteroplasmy is not related to ovarian stimulation but is correlating with the maternal age.

The obtained results are very important, and focus on an important aspect of ART safety. However, the results are still preliminary and very fundamental decisions should not be made as a result of this study, as it could give a false signal to society in a very important aspect of healthcare. Therefore, the manuscript should undergo a major revision before publication.

The following aspects of the study must be clarified:

1) The main drawback of the study is that it was conducted in only one cohort. In order to draw such an important conclusion, the results of the study should be replicated in at least one independent cohort.

2) The study used a wide variety of DNA sources for mitochondrial heteroplasmy studies, like blood samples from newborns and young adults, placenta, buccal cells, and saliva from children and young adults. As using different samples could bias the heteroplasmy analysis, this shortcoming of the study needs to be emphasized and discussed more. If similar samples were used, the result could be different.

3) What was the proportion of heteroplasmy in the examined DNA samples, i.e., proportion of mutated DNA when compared to the wild-type DNA. Can this be analysed? Is it different between ART and SC children?

4) Has the heteroplasmy analysis NGS method also been validated with an alternative method? The authors provided the references for the validation, but was the validation also conducted in the author's lab?

5) It is not clearly mentioned what kind of ART was used, whether IVF or ICSI procedure and what day of embryo development the embryo transfer was made, i.e., early cleavage vs blastocyst stage embryo transfer. This information should also be considered in the analysis.

6) In the discussion, the effect of IVF embryo culture media from different producers/vendors on the birth weight of IVF children is discussed. The reader should be more clearly warned that the given conclusions are based on only a single study and very clear conclusions cannot be drawn from it yet.

7) The low birth weight of ART children compared to SC children is justified just using a single reference, that indicates the lower birth weight for ART children. However, this topic has been the subject of very active research for decades, and the results have been contradictory so far. Therefore, this topic should be discussed somewhat more openly and emphasizing the inconsistency of the results so far.

Dear reviewers,

We would like to thank you for the constructive reviews of our manuscript.

We have carefully revised the manuscript and taken on board all your comments. We have made entirely new figures to replace the previous ones, the text has been thoroughly edited and much information has been added to both the main text, figures and supplemental data.

Based on your comments, we have reanalyzed the dataset to include only the non-synonymous and rRNA variants in the models to predict birthweight. This change resulted in an adjustment in the inclusion of ART individuals, based on the culture medium they were exposed to. Consequently, we have redone all regressions and discriminant analysis. Importantly, these adjustments have not changed the core message of the manuscript on the association of mtDNA variants with birthweight. Reanalysis of the mother-child pairs and oocyte data has revealed that ART individuals carry more *de novo* non-synonymous changes than their spontaneously conceived peers, and that this not only correlates with maternal age, but moreover that the size of the oocyte cohort obtained after ovarian stimulation correlates to the appearance of *de novo* non-synonymous and rRNA variants. In line with all this, we have adjusted the title of the manuscript.

Below you can find a point-by-point answer to all your comments and queries. All changes are highlighted in red in the manuscript.

We hope you agree with us that the manuscript has gained in clarity and its message has been strengthened after this revision.

We look forward to your comments.

Reviewer #1

This article is exploring the effect of heteroplasmic mitochondrial variants load on the birth weight of children born using assisted reproductive technology (ART). The gap in knowledge the authors are attempting to close is important to address given the increase in use of ART. Overall the article is well structured but lacks clarity and there are methodological issues.

Major points

1. It is unclear from the article what the authors aim to test. Are they testing whether heteroplasmic load is associated with birth weight, adjusting for confounders, or are they trying to build a prediction model that could be used for diagnostic purposes. There is also little explanation on why switch to discriminant analyses other than negative results from the logistic model. For example, it would have been helpful to show power calculations in order to judge whether the logistic regression would have been able to detect an effect in the first place.

Our aim was to test the hypothesis that differences in mitochondrial DNA genotype could explain the differences in birthweight observed between ART and SC individuals. For this, we tested if a) there were differences in the mtDNA genotype of ART and SC individuals, and b) if these differences (if present) in mtDNA genotype associate to birthweight.

When testing the association between birthweight and mtDNA variants, we first used a logistic regression, as it is a common approach to test associations while factoring in confounders. However, despite identifying statistically significant effects, the model as such was not efficient at correctly classifying the individuals. For the birthweights below P10, only 1 out of the 13 individuals were correctly classified, this was 4 out of 42 for <P25. This prompted us to test if the discriminant analysis was better suited to develop a model that correctly categorized the individuals, using the same variables as input, which it was. We believe that the differences between the two models do not relate to sample size as such, but to the different approach of the two models to predict whether a case will belong to one group or another.

The discriminant analysis tries to separate two normal distributions in a multi-variate space such that in that multi-variate space the separable groups must exhibit a typical elliptic scatter where two ellipsoids are separated by a linear hyperplane. Although a linear function is also used to discriminate two groups in logistic regression, its philosophy is different as it assumes the existence of a latent score whose zero crossing discriminates between one or the other group. The score is assumed a logistic random variable. Typically the logistic regression is the default choice because it is far more universal as it does not rely on the multivariate normality of the data. However, the latent score needs to be sampled sufficiently dense. Following Vittinghof's 1 in 10 or relaxed 1 in 5 rule, one needs to have 5 observed events per degree of freedom. For instance, in the analysis of the individuals born under P10, the number of events only consists of 17 out of 282 cases which only allows for 3 degrees of freedom. Thus the maximum likelihood estimation for the logistic regression was unable to determine the coefficients. The discriminant analysis, although its more restrictive assumptions, can work for smaller sample sizes as it only requires to determine the covariance of the data and the group means.

We have adjusted the manuscript to more clearly state the hypothesis we tested (lines 119-144) and to provide more detail on the regression and discriminant analysis (lines 375-411).

2. Could the authors give a more detailed description of the discriminant analyses they did. For example, how they decided on $p < 0.2$ for inclusion of variables in the model and how was the cross-validation done. Also, although categorical variables can be used in discriminant analyses, this is not ideal as it violates the normality condition and could cause instability. Further to this, it would help to show what is the added value, if any, of the genetic variants. It would help if the authors report results of discriminant analyses that do not include the genetic variants.

The minimum inclusion criterion is $p < 0.2$ which is a standard choice in a sequential forward regression. Variables which are less significant than 0.2 are not considered to avoid overfitting.

Indeed, discriminant analysis is a prediction model such that its performance is evaluated through a cross-validation. Here a leave-one-out cross validation is applied due to the small sample sizes. The LDA is performed iteratively by using $N-1$ datapoints where the sample left out is validated. This is conducted for all the samples. Although other cross-validation methods exist, these typically expect a large sample size.

Although the inclusion of categorical variables in an LDA approach is somewhat against the basic assumption of multivariate normality, as long as the data projected on the line through both group medians follow a bi-variate normal distribution, the discriminant analysis procedure is sufficiently robust. As aforementioned, the logistic regression approach was inapplicable.

We agree that it is useful to report on the models without the mtDNA variants. We now include in tables 3 and 4 the discriminant analysis (DA) done only on confounding maternal factors, only on mtDNA factors and the combination of mtDNA and confounding factors. We have elaborated in the text, lines 404-411, on the impact the mtDNA has on the models:

“The models to predict the 10th and 25th birthweight percentiles showed both that while the pregnancy-related factors on their own were very good predictors of the birthweight, the addition of the mtDNA factor increased the ability of the model to correctly identify the individuals under the 10th and 25th birthweight percentile. For the <P10 model, the addition of the mtDNA factors increased the accuracy for specifically identifying the <P10 individuals from 40% to 70%. For the 25th percentile, the accuracy increased from 58.1% to 67.7%. Remarkably, the mtDNA factors on their own were more predictive for a birthweight under the 25th percentile than the pregnancy-related factors (62.5% vs 64.9% correct classification respectively).”

3. Given that quantitative traits afford better power, it may be worth as an initial model to use birth weight as a quantitative trait rather than dividing the weight in categories.

We agree with the reviewer. In one of the analyses that were not included in the paper, we tested the use of the Z-score (normalized birth weight for gestational age and sex), but linear regression only identified maternal age and pregnancy hypertension as significantly impacting birthweight. The mitochondrial factors, while negatively impacting the Z-score, did not reach statistical significance. This DA model is not significant and does not predict the Z-score. We therefore choose to not include it in the manuscript.

4. The authors should clearly define what they mean by pathogenic (e.g. causing known mitochondrial conditions versus variant increasing the risk of being born with lower birth weight), especially since they are looking at a complex trait.

In the paper we define potential pathogenicity based on *in silico* predictions: lines 226-229 “The pathogenicity scores were obtained through *in silico* analysis using MutPred2 and MitoTip for the non-synonymous and tRNA variants respectively. MutPred2 scores >0.61 and MitoTip scores >50% were considered as potentially pathogenic.”

In the manuscript lines 258-260 we mentioned that “we found one SC individual with a proven pathogenic tRNA-coding variant at homoplasmic level (mt.14674T>C, associated with Reversible Infantile Respiratory Chain Deficiency(Schon et al., 2012))”.

And in lines 277-281: “Four individuals (3 ART and 1 SC individuals) carried a proven pathogenic heteroplasmic tRNA-coding variant, including the variants m.3243A>G and m.5521G>A (both associated with the MELAS/MERRF syndrome(Herrero-Martí-N et al., 2010; Li et al., 2022)), and m.5703G>A (associated with mitochondrial myopathy and MERRF(Fu et al., 2019)).”

For the variants without proven association to a phenotype, while we explored their potential pathogenicity with the *in silico* predictions and included them in some of the supplementary analysis, we do not further utilize these in the paper because of the lack of certainty of their effect. We have added this in the new section on limitations of the study, in the discussion, lines 555-568.

5. In relationship to the above and since common mitochondrial variant have been associated with BMI and body composition parameters in adults, the authors should specify whether there was any difference between ART and SC children in the distribution of such previously associated variants.

We searched the literature and found the following (additional) publications reporting a link between mtDNA variants and body composition. We apologize in advance in case we have missed important studies and kindly ask the reviewer to point them out to us if needed. We have queried our dataset and found none of these variants back in a consistent manner or associating to birthweight, with the exception of haplogroup T. We have included this in lines 308-325 of the results and lines 494-507 of the discussion.

We have included the following papers:

(Yang et al., 2011) The authors identify mt4823 (in MT-ND2) and mt8873 (in MT-ATP6) as associated to reduced fat mass values. These variants are not present in our dataset as heteroplasmic variants nor as homoplasmic variants outside of the haplogroups.

(Flaquer et al., 2014) The authors find significant results for variants in MT-CO1, MT-CO3, MT-ND1, MT-ND2 and MT-ND4L. None of the identified variants are found in our dataset. Interestingly, 3 of the 5 variants they report are synonymous.

(Knoll et al., 2014) In the discovery phase, the authors found association with obesity for the frequent allele G of m.8994G/A located in ATP6. Haplogroup W was nominally overrepresented in the controls. These findings could not be confirmed independently. The D-loop variants m.16292C/T, $p = 0.007$, m.16189T/C, $p = 0.048$ also associated to obesity. m.8994G can be found in homoplasmic state in one of our samples and in an oocyte of a stimulated ovarian cycle as a heteroplasmic. We have no birthweight data associated to these samples. m.16292T appears as a heteroplasmic in one ART child and their mother. The child had birthweight >P25.

(Casteels et al., 1999; Parker et al., 2005) find that m.16189T associates to thinness at birth. In our cohort, 57 individuals carry this variant as a homoplasmic (often as part of their haplogroup) or as a heteroplasmic. It does not associate to <P10 ($p=1$), <P25 ($p=0.851$) or <P10 in SC and Vitrolife (<P10, 0/17, >P10 38/265, Chi-square $p=0.141$) and <P25 in SC and Vitrolife (<P25 4/48, >P25 34/234, $p=0.354$). These results did not change in terms of significance if considering only the variant when it is present outside of the haplogroup or as a heteroplasmic.

(Ebner et al., 2015; Nardelli et al., 2013) haplogroup T is found to be associated with obesity. In our cohort it is underrepresented in the <P25 cohort. In the logistic regression model for <P25, it does not reach statistical significance. In both the logistic regression and the discriminant analysis it has a negative effect on the probability of being <P25.

6. It would be helpful to add a paragraph discussing the limitations to the study in the discussion section.

We fully agree. We have added a section on limitations of the study in lines 555-568, covering the following points:

- 1) the study cohort is in fact composed of two cohorts of samples that when analyzed on their own do not reach statistically significant results
- 2) our final study cohort shows no significant differences in birthweight between ART and SC individuals
- 3) the associations between haplogroups and other homoplasmic variants and birthweight are likely underpowered and should be taken with caution
- 4) future replication studies with larger cohorts will be key to validate our findings
- 5) the variants we report on have no proven pathogenic effect

Minor points

1. Is it possible for the authors to call mitochondrial copy number from the data they have. It would be interesting to see if there are any difference between ART and SC children and whether copy number affects birth weight.

The method we used did not allow us to establish copy number. At the start of the study, we had some concerns

that differences in copy numbers could bias our sequencing results, and also considered the possibility that there would be differences between ART and SC individuals. We ran mtDNA copy-number assays on a subset of 83 samples, using qPCR. We compared the blood samples from ART and SC individuals using two probes for the mitochondrial genome and one probe for the nuclear genome. We averaged the copy numbers obtained from the two assays and compared ART to SC individuals and found no significant differences between groups (Mann-Whitney U test, $p=0.232$), suggesting that at least there were no large differences in mtDNA copy number between the groups that could bias our genotyping. We did not pursue this further, as it appeared unlikely that we would reliably draw any further conclusions because of the different sample types we were working with (blood, buccal cells and placental tissue).

2. It would be helpful if the authors used a clearer way to represent the results in figure 1d. The figure is difficult to read.

We agree that this panel was not optimal. We have adjusted all figures in the paper, added panels and made a new figure with the overview of all variants that we think is an improvement over the original one.

3. Line 353 change “are predictive for” to “associated with”

Done

4. In the section exploratory factor analyses within supplementary methods, could the author add a figure showing how the eight variables used in the analyses were distributed and also how they correlate with each other. That would help to judge the normality and linearity of the data.

We have added a hypercube in Figure 1L to illustrate the co-variance of the variables and the correlation matrix of the factor analysis can be found in the supplementary data. In figure 1J we have also plotted the variables, which are not following a normal distribution, and the factor scores (figure 1M), which are normally distributed (Kolmogorov-Smirnov $p < 0.001$)

5. Could the authors please re-format the supplementary data tables. They are very difficult to follow.

We have revised all the supplemental material and structured it to follow the figures and tables in the main text.

Reviewer #2

In this manuscript, Mertens et al analyze mtDNA sequencing data from 270 individuals born via assisted reproductive technologies (ART) and 181 from spontaneous conception (SC), as well as from 157 mother-child pairs and 113 oocytes from cycles with or without ovarian stimulation (OS). For each individual, they compute the sum of heteroplasmy level for eight variant categories, and use exploratory factor analysis to identify four factors that represent the majority of variance across the eight categories. Four factor scores are then calculated for each sample, using the sum of the heteroplasmy level across categories, which are used for majority of the analyses and significant conclusions.

The authors identify significant differences in mtDNA variation between groups. They identified higher factor 2 scores in ART than SC individuals, and in SC individuals who were P<10 or P<25 birth weight. They also note SC P<25 individuals more frequently carried variants in proteins and rRNAs, and that SC and female ART individuals P<25 showed a higher cumulative heteroplasmic load of these variants. They also reported significantly higher factor 2 scores in ART individuals for transmitted but not de novo variants. They conclude that individuals born after ART more frequently carried protein and rRNA variants with higher loads, and that these differences were predictive of birth weight.

Overall, the hypothesis is interesting, and a link between mtDNA variation and birth weight in ART and SC individuals would be noteworthy and significant. However, many of the major results of this study require additional evidence to support the conclusions made.

Major:

Several conclusions are based solely on the factor scores, which have no validated utility for assessing the impact of mtDNA variants. Rather, these factor scores apply somewhat arbitrary weights to variant categories.

The factor scores were meant mainly to be used to explore the data and to identify the main components of diversity across individuals (in figure 1) or oocyte samples (in figure 3). Conversely, the factor analysis results are not strictly necessary to prove the associations seen in figures 2 and 3. In figure 2, the association between variants and birthweight is supported by the categorical value 'presence of non-synonymous protein coding and rRNA variants' and by the sum of heteroplasmic loads of non-synonymous protein coding and rRNA variants (now in figure 2l, 2m and 2n), and the factor scores are not part of the logistic regression and discriminant analysis. The same holds true for figure 3, where the factor analysis is a means to discovery of a different incidence of *de novo* protein coding variants in ART children, but the presence of certain variants or their heteroplasmic loads are the basis of further analysis.

In this version of the manuscript, we have added more information to the figures and text to clarify this and further support the findings (for instance lines 295-302, 328-336)

Many conclusions using factor 2 scores rely on the inclusion of synonymous (i.e. under 'protein variants'). Synonymous variants do not change protein sequence and can be regarded as evidence of benign impact per clinical variant classification guidelines (e.g. PMID: 32906214). Significance tests which compare different frequencies or heteroplasmy levels for protein and rRNA variants between groups should be repeated with synonymous excluded (i.e. limit to non-synonymous and rRNA). This would help strengthen the claim that rRNA and protein variants are driving the association with birth weight through impacts on mitochondrial function.

Synonymous variants are indeed benign and we were undecided on whether to include them or not in the analysis. The main reason for keeping them in was the fact that we found a number of papers providing evidence for an effect on codon usage, which could modulate mitochondrial function (Jia and Higgs, 2008; Shtolz and Mishmar, 2023). We agree with the reviewer that it would strengthen our claim to include only the non-synonymous changes, so we have done the analysis again taking only rRNA and non-synonymous variants into account. Reassuringly, the overall outcomes of the study remain the same. We have made a new figure 2, as well as adjusted large sections of the manuscript (lines 304-411).

Can any other evidence be provided, such as population frequency data, to strengthen the conclusions that protein and rRNA variants are driving low birth weight through functional effects?

This is an interesting suggestion. The most comprehensive study of mitochondrial DNA diversity we could find is the work by Patrick Chinnery's group published in Science in 2019. They studied mitochondrial heteroplasmy in 12,975 whole-genome sequences belonging to 1526 mother-offspring pairs. They found that 45.1% of individuals carry heteroplasmic variants, 30% of which are in protein coding and/or rRNA regions (Wei et al.,

2019). This frequency is relatively close to 25% - as in the 25th percentile of birthweight. The data of Wei et al does not allow for easily extracting whether the variants were synonymous or non-synonymous. In our dataset, 40% of individuals have protein coding and/or rRNA variants, 30% having non-synonymous and/or rRNA variants. Assuming that the ration syn/nonsyn will be about the same, if we extrapolate to the Wei data, approximately 22% of individuals would carry non-synonymous and/or rRNA variants. While this is indeed suggestively close to the lower birthweight percentile, it does not provide evidence for a link, and we are unsure whether this should be included in the paper. We are open to other ideas and suggestions by the reviewer. Certainly, this calculation could be done by the authors of the Science paper, or alternatively a study could be initiated to test the hypothesis.

Also, why are tRNA variants systematically excluded from most analyses?

In the first version of the manuscript, both the heteroplasmic and homoplasmic tRNA variants were included in the univariate analysis and were only included in further modelling when $p < 0.2$. In the current version of the manuscript, we have included the sum of heteroplasmic loads for tRNAs in figure 2e (but their presence or load does not associate to the birthweight percentiles), and the homoplasmic variants are in figure 2c and figure 2k. The homoplasmic variants were included in the forward binary logistic regression and discriminant analysis models for $<P10$ and $<P25$ (tables 2, 3 and 4), and all variants (including heteroplasmic tRNA variants) were included in backward conditional binary logistic regressions. In these, the tRNA variants were not retained in the model, suggesting that they do not associate to birthweight percentile.

The authors state their motivation was to investigate the impact of mtDNA variation on the increased risk of lower birth weight in ART vs SC, however their cohort shows no significant difference in $P < 10$ and $P < 25$ categories between the ART and SC groups, per Table 1. How does this impact their major conclusion that mtDNA variants drive low birth weight and that this is more common in ART individuals?

The reviewer is absolutely right. Once the study cohort was complete, we found that there were no differences in birthweight. It does seem likely that if there would have been significant differences, we would have had more pronounced differences in the mtDNA variation. This is a weakness of the study and is now discussed in the paragraph dedicated to the limitations of this work (lines 545-558). However, we do believe that demonstrating a correlation between mtDNA variants and low birthweight is a significant finding in its own right.

Lines 233-236: "This indicates that individuals born after ART tend to carry higher loads of heteroplasmic variants in the protein-coding and/or rRNA regions" –the authors should provide support for this conclusion drawn from the factor scores by showing the distribution of the heteroplasmy level of variants in each category, and statistical assessment of their difference between ART and SC. In Table S6 the proportion of heteroplasmic variants in rRNA genes is very similar between ART and SC (8.7% vs 8.0%), implying that any impact must be in their heteroplasmy level. However most rRNA variants detected appear to be $<20\%$ heteroplasmy level per Figure 1d, a mutant load at which functional impacts are particularly unclear. The proportion of ART and SC individuals with heteroplasmy, across variant categories, should also be shown as done for homoplasmy in Figure 1b and Table S5.

The complete sentence in line 233-236 was "This indicates that individuals born after ART tend to carry higher loads of heteroplasmic variants in the protein-coding and/or rRNA regions, in combination with lower or similar heteroplasmic loads in the other regions compared to their SC peers." The factor analysis provided an indication of differences in combinations of variants, while there were no statistically significant differences at the level of the variant itself. In this case, the score of factor 2 is driven towards positive values by the presence of variants in protein coding regions and rRNA (most strongly by non-synonymous variants (0.713) but also by synonymous (0.356) and rRNA (0.554)), and, at the same time, negatively by variants in the other regions, predominantly OHR and TAS. So the absence of variants in those regions, at the same time as having more variants in the protein coding and rRNA, are resulting in the higher factor 2. In the revised version, we have also added more information to further clarify this aspect of the analysis (lines 295-302).

In the current version of the manuscript, figure 1h shows the number of variants of different types carried by individuals in the ART and SC groups. There are no statistically significant differences, although the distribution of non-synonymous variants has a Chi-square with $p = 0.067$, where there are 16.6% of individuals in the ART group carrying 1 non-synonymous variant (vs 13.8% in SC) and 3.7% of individuals carrying 2 or 3 variants (vs 0.5% in SC). Figure 1j now shows the heteroplasmic loads of the variants across locations and groups, with no statistically significant differences between the groups. Interestingly, the mean load of heteroplasmic variants in the HV region is lower in ART individuals (ART: 4.55%, SC: 7.02%, Mann-Whitney U test $p = 0.318$). The

combination of the higher number of non-synonymous variants and the lower loads in HV are in line with higher factor 2 scores, illustrating how this method can extract significant variation from variables that do not show statistically significant differences after univariate analysis.

- Table 1 lists a range of tissue types analyzed. Did the ART and SC individuals have multiple tissues analyzed? Was the distribution of sample types used for DNA similar between ART and SC? Given sample type can influence heteroplasmy, it would be important to ensure results are not driven by sample type differences between groups.

We did not test multiple tissues per individual in the ART and SC cohort, but we did so for the oocyte donors. We had buccal swab and blood from 24 donors, 13 of which carried heteroplasmic variants in at least one of the two samples, 24 variants in total (see table below). Fourteen of these variants matched between tissues, 10 appeared in only one of the tissues, 7 in buccal swab in 3 in the blood. Typically, mismatches were for variants with low heteroplasmic loads. Overall, the sample size is too small to draw conclusions.

Donor	Position	Variant	Buccal	Blood	
DONOR 10	9525	G>A	MT-CO3, A107T	1.51	3.97
	2532	T>C	MT-RNR2	1.75	2.43
	215	A>G	HV;OHR	3.2	0
	203	G>A	HV;OHR	37.35	5.32
DONOR 14	2906	InsCT	MT-RNR2	100	50.3
DONOR 15	16093	T>C	HV	93.6	98.5
DONOR 16	14619	A>G	MT-ND6, F19L	0	2.18
	7598	G>A	MT-CO2, A5T	96.04	95.32
DONOR 17	15616	C>T	MT-CYB, Syn	14.62	12.2
	11812	A>G	MT-ND4, Syn	86.01	87.01
DONOR 19	199	T>C	HV;OHR	2.59	
DONOR 02	12773	G>T	MT-ND5, G146V	0	2.1
	13761	A>C	MT-ND5, Syn	0	5.1
	8783	G>A	MT-ATP6, G86E	1.66	0
	195	C>T	HV;OHR	94.25	58.64
DONOR 20	16129	G>A	HV	0	2.78
	15597	T>C	MT-CYB, V284A	1.6	0
	15239	T>C	MT-CYB, W165R	2.28	0
	146	T>C	HV;OHR	29.14	37.08
	15757	A>G	MT-CYB, Syn	81.31	82.13
DONOR 21	16270	C>T	HV	55.13	73.37
DONOR 24	15043	G>A	MT-CYB, Syn	13.3	15.64
DONOR 03	152	T>C	HV;OHR	1.6	0
DONOR 06	214	A>G	HV;OHR	8.03	0
DONOR 07	455	InsTTTC	NonCod	91.4	13.8

With regards to controlling the samples for tissue of origin, the distribution of sample types was not the same between ART and SC. In the previous version of the manuscript, we did control if the samples differed in their heteroplasmic variant burden, originally by using the factor analysis (supplementary figures S1 in the first version of the manuscript, no statistically significant differences were detected). In this revised version, we have included the plots for total heteroplasmic load in the three types of tissues and across the three age categories in figure 1F and 1G. No statistically significant differences were found.

- Lines 245-249: “Conversely, SC individuals who were P10 or >P25... In line with this, SC <P25 individuals more frequently carried variants in protein and rRNA loci” – the authors should provide data on the proportion of SC individuals carrying heteroplasmies for each variant categories to support this, as Table S11 referred to does not appear to show this. For Table S11 it would be helpful to state in the legend exactly what the % and () represent and how those percentage values were calculated.

We agree that we should have provided more data on this aspect, and our apologies for the missing table.

We have made a new figure 2, including more information. Figures 2e, 2f, 2g, 2l, 2m and 2n now contain data on the proportions of individuals carrying each type of variant and their heteroplasmic loads, as well as tables S10 to S23. We have restructured the supplementary data, and added explanations of how percentages are calculated in all tables.

- Line 259-263: “...male individuals that were subjected to Cook and UZB medium (e.i. an in-house medium made in the UZ Brussel) are more frequently born with a birth weight <P25 than females (25.5% in males vs 10.7% in females). Remarkably, only 7.7% of these <P25 males carried variants in protein- and rRNA-coding loci as compared to 30.0% of the <P25 females” – the authors should provide the data underlying this observation (i.e. supplementary table), as it forms the basis of excluding male ART individuals from all subsequent analyses. Statistical analysis on the difference in birth weights between males and females across all culture mediums used would also be of interest to support the effect of the media on males and their exclusion.

We agree that this should be included in the paper, with the necessary caution because the sample sizes are relatively small after subdividing by culture medium and sex, and after adjusting the models to include only non-synonymous and rRNA changes. There are two main observations. First, males appear more susceptible to having a lower birthweight than females when subjected to Cook medium (this was reported in the study by Nelissen et al (Nelissen et al., 2012) and this same trend appears true for UZB medium. Secondly, having been exposed to UZB and Cook has a higher impact on birthweight than mtDNA variation. We have now included all this in figure 2h, and in the manuscript, lines 337-374.

In the previous version of the manuscript, we had removed the males from the model as they appeared to be more influenced by the culture media than females. With the current adjustments, especially by only considering non-synonymous and rRNA changes, we feel it is more accurate to remove all individuals in UZB and Cook medium when factoring in mtDNA variance into predicting birth weight.

We have reanalyzed the data from that perspective, and made the logistic regressions considering only the SC individuals and the ART children born after the use of Vitrolife (which incidentally were the largest group of the study). The outcomes are not drastically different from considering only females in ART, as in the previous version, but we feel they are more accurate. We also have included the culture media aspect in the discussion and added it to the weakness of the study.

- The discriminant analysis showing mtDNA variants in combination with other data could predict birth weight category is a particularly interesting result. Does the classification precision decrease when mtDNA variants are removed from the model, i.e. how much of this is related to mtDNA vs factors such as maternal age, hypertension and or smoking? This is important for their assertion that “differences in the mitochondrial genome were also predictive of the risk of a lower birth weight percentile”. For assessment of confounding factors/parameters for the logistic model, listed in Table S12, it appears that assessment of homoplasmies was restricted to tRNA and heteroplasmies to protein plus rRNA. The authors should provide the values for all categories of homoplasmies and heteroplasmies (i.e. per Table S11) or provide a sound justification for their exclusion from the logistic regression models.

We agree that for the discriminant analysis it is important to show the effect of the mtDNA factors in the model. For this, we have now generated the models using only the maternal factors, only the mitochondrial factors and the combination of both (tables 3 and 4). These results show that the mtDNA factors add to the ability of the model to identify individuals with lower birthweight.

We have elaborated on the impact the mtDNA has on the models in the text (lines 400-411): “The models to predict the 10th and 25th birthweight percentiles showed both that while the pregnancy-related factors on their own were very good predictors of the birthweight, the addition of the mtDNA factor increased the ability of the model to correctly identify the individuals under the 10th and 25th birthweight percentile. For the <P10 model, the addition of the mtDNA factors increased the accuracy for specifically identifying the <P10 individuals from 40% to 70%. For the 25th percentile, the accuracy increased from 58.1% to 67.7%. Remarkably, the mtDNA factors on their own were more predictive for a birthweight under the 25th percentile than the pregnancy related factors (62.5% vs 64.9% correct classification respectively).”

For the logistic regression, we have made a new table that now shows all the homoplasmies and heteroplasmies and their univariate analysis p-values (Table S24). We have now carried out the logistic regression with two approaches, to contrast the outcomes. First, we used the forward regression with the variables of Table S24 with a $p < 0.2$. For the heteroplasmies, we choose to use the non-syn+rRNA variable, and excluded the other variables on heteroplasmies because they overlap with the non-syn+rRNA variable. In this model, the minimum inclusion criterion of $p < 0.2$ was used to avoid overfitting. Second, we carried out the binary logistic regression with a backward conditional approach, with all variables listed in table S24 as input.

The two approaches identify smoking, pregnancy hypertension and homoplasmic tRNA variants as predictive for a birthweight under the 10th percentile, and maternal age and the presence of non-synonymous protein coding and rRNA variants as predictive for the birth under the 25th percentile. The effect of the haplogroups is variable, and they are not consistently statistically significant (Tables S25 and S26). We have added discussed this in the strengths and weaknesses in the discussion lines 555-568.

- Lines 292-303: “We found a higher incidence of ART mother-child pairs presenting de novo variants in the protein-coding regions....The scores for factor 2 (driven by the presence of protein-coding and rRNA variants) were significantly higher in ART individuals for the transmitted but not for the de novo variants ... These findings suggest that the differences in mtDNA variants between ART and SC children are due to a higher transmission of protein- and rRNA-coding variants” - The authors should provide support for this conclusion drawn from the factor scores by showing data on the transmission and de novo origin of variants across each category for ART and SC individuals (i.e. proportion of individuals with said variants and differences in their heteroplasmies level, especially for transmitted where no data is presented), and statistical assessment of any difference between groups. How do they contextualize the higher incidence of protein de novo in ART with the factor 2 scores only being significant for transmitted but not de novo?

We agree that the factor analysis is rather confusing in this part. We have removed it altogether from the oocyte analysis, and have adjusted it in other parts of the manuscript. We have made a new figure 3 with more data, a table 5 including additional analysis on the number of variants, as well as supplementary data Table S27 including all the transmitted variants and their loads, and Table S28 including all *de novo* variants and their loads.

- Lines 317-319: “No trends were observed in changes in heteroplasmic load of the same variant in oocytes obtained after a natural and an OS cycle (Figure S3 in supplementary appendix).” – it appears that heteroplasmic load was more likely to decrease in oocytes from natural vs OS cycle (48.5% vs 31% in Figure S3). Can the authors explain why they regarded this data to represent no change?

We tested this distribution using a Chi-Square test and found it not to be significant. The number of variants is also very small, which may explain the lack of significance of these differences. We included all the transmitted variants in figure 3 and table S29 and S31, and the Chi-square in line 451.

- Lines 331-333: “the location of the de novo variants as well as the number of oocytes carrying de novo protein-coding and rRNA variants, was similar between the natural and OS cycle oocytes (Figure 3d)” – Figure 3d shows approximately two-fold increased rRNA and non-synonymous de novo variants in OS vs natural, was statistical testing used to conclude they were similar?

Yes, we tested this by Fisher exact (14.5% vs 23%, $p=0.338$), we should have mentioned it in the text. We have now added new panels to figure 3, and explicitly state the statistical tests, but the conclusion remains that there are no significant differences between the oocytes based on them being from a natural cycle or a stimulated cycle. Conversely, we find that oocytes from larger cohorts of retrieved oocytes do carry higher numbers of non-synonymous and rRNA variants. This is now part of figure 3 and Table 5, and text lines 462-480.

- Lines 347-350: “the maternal age was significantly higher in the ART group (table I, t-test, $p < 0.0001$), potentially explaining why ART children have a higher incidence of de novo variants”- The authors do not present

data showing that there is a higher overall de novo rate in ART vs SC individuals – this should be provided to support their assertion.

In this version of the manuscript, we include new analyses and data that show that ART individuals have higher rates of non-synonymous and rRNA variants (figure 3c and 3d), and that while maternal age increases the overall rate of *de novo* variants in children and oocytes, the size of the cohort of oocytes retrieved correlates to the number of non-synonymous and rRNA variants found in them. This suggests that in ART individuals, potentially a combination of maternal age and of the procedure of ovarian stimulation lays at the basis of a higher number of non-synonymous and rRNA variants. SC individuals born from older mothers carry overall more de novo variants, therefore increasing the risk of carrying also more de novo non-synonymous and rRNA variants. It is interesting to note that the maternal age is an important factor associated to birthweight, and it seems plausible that this occurs by a double effect: an increase in the mtDNA variants and a less optimal environment for fetal growth.

- Lines 354-355: “the presence of protein-coding and rRNA mtDNA variants was predictive for the birth weight percentile of the females but not the males” – can the authors point to their data showing the assessment of variant predictive value in males vs female birth weight? The discriminant analysis used to classify individuals (Figure 2j-k) appeared to only be performed on females.

In the current version of the manuscript we have decided to include only non-synonymous and rRNA changes. With this adjustment, we also found that it is more correct to remove the individuals that were subjected to UZB and Cook medium when factoring in mtDNA variance into predicting birth weight.

We have reanalyzed the data from that perspective, and made the logistic regressions considering only the SC individuals and the ART children born after the use of Vitrolife (which incidentally were the largest group of the study). All the data is now shown in the manuscript. The outcomes are not drastically different from considering only females in ART, as in the previous version, but we believe are more accurate. We also have included the culture media in the discussion, particularly in the weakness of the study lines 555-568.

Minor:

- The method used to determine haplogroups, and haplogroup variants, should be stated.

The haplogroups were determined by haplogrep (<https://mitoverse.i-med.ac.at/>). This has been included in the materials and methods, lines 214.

- More than 1kb of the total mtDNA is excluded from analysis, this should be reiterated in the results text and highlighted in Figure 1d.

We included a schematic overview of the regions that are included/excluded (text lines 221-223, figure 1A and supplementary table S2)

- Do the authors have a rationale for why they segment intergenic variants into four groups (HV, non-coding, OHR and TAS)? This would appear to reduce power for analyses across intergenic variants.

We decided to categorize the variants intergenic groups as they may have different functional impacts. Considering the comment of the reviewer, we have done an analysis that pools all intergenic variants together. There are no statistically significant differences between ART and SC individuals in terms of total heteroplasmic loads in these regions ($p=0.138$, see matrix below). The first dimension is still dominated by the presence of protein coding variants, the second by noncoding variants and the third by tRNA variants. The rRNA variants are now mostly representing a negative score in factor 3. Overall, we don't find that this analysis is improving on the previous one, and have chosen to keep the original factor analysis in figure 1.

Rotated Component Matrix^a

	Component		
	1	2	3
HETCL.Syn	.759	.236	-.136
HETCL.NonSyn	.729	-.227	.110
HETCL.rRNA	.154	-.321	-.569
HETCL.tRNA	.105	-.164	.821
HETCL.NonCodingAll	.022	.892	.022

Extraction Method: Principal Component Analysis.
Rotation Method: Varimax with Kaiser Normalization.

a. Rotation converged in 5 iterations.

- Line 215 “individuals frequently carried multiple variants with different heteroplasmic loads” appears at odds with the average of ~1 heteroplasmic variant per individual reported. Can the authors clarify?

We agree that this statement was not clear. We have replaced it with “60.4% of ART and 61.9% of SC individuals carried up to 5 heteroplasmic variants per person” (lines 269-270) and made new panels in figure 1 to show the number of variants the individuals carry for each category. Table S6 shows the numbers of individuals carrying different amounts of variants in each of the categories, and the p-values for the Chi-square tests.

- It should be remarked whether the statistical tests applied were one or two sided, as appropriate.

All statistical tests were two-sided. We have added this in the methods, line 241-242.

- In general the factor score dot plots are hard to compare, could median or mean bars be added or the jitter increased (to avoid overlap of points). What does the horizontal line represent?

We agree with the reviewer that these plots were not clear enough. We have made new figures that we think are clearer. The horizontal lines in the scatter plots represent the mean, we have added this to the legends of the figures.

- Lines 322-324: “To determine whether the de novo variants identified in multiple oocytes of the same donor could be considered as independent events in the statistical analysis” – do the authors mean different de novo mutations? As if the same de novo variant, then wouldn’t identification in multiple oocytes from the same donor be consistent with it not being a true de novo but rather under the limit of heteroplasmy detection?

We agree that this is confusing. We wanted to test if oocytes of one specific donor would be more prone to a specific pattern in *de novo* variants. For this, we carried out a factor analysis on the *de novo* variants of all oocytes (*de novo* defined as detected in only one oocyte of the donor) and correlated the factor scores of the oocytes of one donor to each other. We hypothesized that if there were patterns in the *de novo* variants of a donor, the factor scores would correlate well with each other. Since they did not, we considered that there were no patterns per donor.

Since we have removed the factor analysis from this part of the manuscript, we have also eliminated this part to not add further confusion.

- Lines 374-375: “The exact mechanisms for the association between non-disease causing mtDNA variants and birth weight are yet unclear” – the authors should clarify in the results text if disease-causing variants were excluded, and how these were defined.

In the paper, we define potential pathogenicity based on *in silico* predictions (mentioned in lines 226-229). The pathogenicity scores were obtained through *in silico* analysis using MutPred2 and MitoTip, for the non-synonymous and tRNA variants respectively. MutPred2 scores >0.61 and MitoTip scores >50% were considered as potentially pathogenic.

In the manuscript we mentioned that we found one SC individual with a proven pathogenic tRNA-coding variant at homoplasmic level (mt.14674T>C, associated with Reversible Infantile Respiratory Chain Deficiency (Schon et al., 2012)) (Lines 258-260). Four individuals (3 ART and 1 SC individuals) carried a proven pathogenic heteroplasmic tRNA-coding variant, including the variants m.3243A>G and m.5521G>A (both associated with the MELAS/MERRF syndrome (Herrero-MartíN et al., 2010; Li et al., 2022)), and m.5703G>A (associated with mitochondrial myopathy and MERRF (Fu et al., 2019)) (Lines 277-280).

We have now also tested for variants that have been reported to be associated to birthweight, as per request of reviewer 1.

We searched the literature and found the following (additional) publications reporting a link between mtDNA variants and body composition. We have queried our dataset and found none of these variants back in a consistent manner or associated to birthweight, with the exception of haplogroup T.

(Yang et al., 2011) The authors identify mt4823 (in MT-ND2) and mt8873 (in MT-ATP6) as associated to reduced fat mass values. These variants are not present in our dataset as heteroplasmic variants nor as homoplasmic variants outside of the haplogroups.

(Flaquer et al., 2014) The authors find significant results for variants in MT-CO1, MT-CO3, MT-ND1, MT-ND2 and MT-ND4L. None of the identified variants are found in our dataset. Interestingly, 3 of the 5 variants they report are synonymous.

(Knoll et al., 2014) In the discovery phase, the authors found association with obesity for the frequent allele G of m.8994G/A located in ATP6. Haplogroup W was nominally overrepresented in the controls. These findings could not be confirmed independently. The D-loop variants m.16292C/T, $p = 0.007$, m.16189T/C, $p = 0.048$ also associated to obesity. m.8994G can be found in one of our samples, as a homoplasmy, and in an oocyte of a stimulated ovarian cycle as a heteroplasmy. We have no birthweight data associated to these samples. m.16292T appears as a heteroplasmy in one ART child and their mother. The child had birthweight >P25.

(Casteels et al., 1999; Parker et al., 2005) find that m.16189T associates to thinness at birth. In our cohort, 57 individuals carry this variant as a homoplasmy (often as part of their haplogroup) or as a heteroplasmy. It does not associate to <P10 ($p=1$), <P25 ($p=0.851$) or <P10 in SC and ART females ($p=0.491$) and <P25 in SC and ART females ($p=0.353$). These results did not change in terms of significance if considering only the variant when it is present outside of the haplogroup or as a heteroplasmy.

(Ebner et al., 2015) haplogroup T is found to be associated to obesity. In our cohort it is underrepresented in the <P25 cohort. In the logistic regression model for <P25, it does not reach statistical significance. In both the logistic regression and the discriminant analysis it has a negative effect on the probability of being <P25.

Reviewer #3

The manuscript by Mertens and her colleagues addresses a very important topic - the health risks of children born following the use of assisted reproduction techniques (ART). The approach of this work is very interesting and complex, using DNA samples from ART and spontaneously conceived (SC) individuals, mother-child pairs, and the study of individual oocytes. Whereas the oocytes were obtained both as a result of a natural and a stimulated cycle. The obtained DNA samples were analyzed for inherited mutations in mitochondrial DNA and for inherited and de novo heteroplasmy.

The results of the study are intriguing, i.e., ART children more frequently carry variants with higher heteroplasmic loads. Heteroplasmy in ART children are either maternally inherited or represent de novo mutations, primarily in mitochondrial genes encoding proteins and rRNA. In addition, heteroplasmy is more common in both ART and SC of small for gestational age children. Moreover, the single oocyte analysis revealed that heteroplasmy is not related to ovarian stimulation but is correlating with the maternal age.

The obtained results are very important, and focus on an important aspect of ART safety. However, the results are still preliminary and very fundamental decisions should not be made as a result of this study, as it could give a false signal to society in a very important aspect of healthcare. Therefore, the manuscript should undergo a major revision before publication.

The following aspects of the study must be clarified:

1) The main drawback of the study is that it was conducted in only one cohort. In order to draw such an important conclusion, the results of the study should be replicated in at least one independent cohort.

We agree with the reviewer that replication will be very important, and should especially be carried out by other research groups to demonstrate reproducibility and soundness of our findings. As such, our work can be considered as an important initial observation with far-reaching consequences if confirmed. Carrying out such a study ourselves for the revision of this manuscript would mean repeating the work that took us several years, even taking into account that the two centres involved recruit from a large patient and ART offspring pool that is fairly unique. We hope that the reviewer understands that this is beyond the scope of the current manuscript. We have discussed the limitations of this work and the importance of replication studies in the discussion lines 555-568.

Conversely, it is worth noting that this study is in fact already composed of two cohorts: one recruited in Brussels and one in Maastricht. When we analyze the two cohorts separately for the main outcomes of this study and implementing the current changes to the manuscript, we see that they both follow the same patterns, albeit they are more pronounced in the Brussels SC individuals. For the Maastricht cohort, 48.3% of the SC and ART individuals exposed to Vitrolife with a birthweight under the 25th percentile carry a nonsynonymous or rRNA variant (vs 35.2% of the individuals >P25, $p=0.210$). For the SC individuals from the Brussels cohort, 57.9% of <P25 individuals carry a nonsynonymous and rRNA variant, vs 18.9% of >P25 ($p<0.001$). Given that the sample sizes for individuals with lower birthweights are limited, pooling the two cohorts was required to obtain statistically significant results.

2) The study used a wide variety of DNA sources for mitochondrial heteroplasmy studies, like blood samples from newborns and young adults, placenta, buccal cells, and saliva from children and young adults. As using different samples could bias the heteroplasmy analysis, this shortcoming of the study needs to be emphasized and discussed more. If similar samples were used, the result could be different.

We fully agree, and the other reviewer also commented on this. We have made new panels in figure 1f and 1g text to address this issue. We found no statistically significant differences across sample types and ages at collection.

3) What was the proportion of heteroplasmy in the examined DNA samples, i.e., proportion of mutated DNA when compared to the wild-type DNA. Can this be analysed? Is it different between ART and SC children?

We are unsure of what the reviewer means. We did measure the heteroplasmic levels of each variant for all samples, we refer to it as heteroplasmic load (i.e. number of mutated DNA copies over the wild type copies, as extrapolated from the number of sequencing reads with the variant vs reads without the variant, as specified in the methods section lines 216-219). We did not measure absolute copy numbers of mtDNA relative to nuclear DNA.

Figures 1 and 3 show the heteroplasmic load for all variants that we detected in the different samples. In figure 1 we have also added panels showing the total number of variants per individual per category, which is not statistically significantly different between ART and SC.

4) Has the heteroplasmy analysis NGS method also been validated with an alternative method? The authors provided the references for the validation, but was the validation also conducted in the author's lab?

The method was setup in our lab in 2017. It was tested on mixes of DNA samples to mimic low-level heteroplasmy and on single cells, and the analysis of the same samples was repeated multiple times with consistent results. The results were published in (Mertens et al., 2019; Zambelli et al., 2017) and the method was used to study bulk DNA samples and single oocytes, blastomeres and stem cells in (Mertens et al., 2022; Zambelli et al., 2018). We have added additional information in the methods section lines 229-233.

5) It is not clearly mentioned what kind of ART was used, whether IVF or ICSI procedure and what day of embryo development the embryo transfer was made, i.e., early cleavage vs blastocyst stage embryo transfer. This information should also be considered in the analysis.

We agree with the reviewer this should have been stated in the manuscript. All individuals were born after fresh embryo transfer at the cleavage-stage. We have details on whether the ART cycle was IVF or ICSI for only 150 of the 270 ART samples: 131 ICSI and 19 IVF. We did not include this information in the analysis because IVF and ICSI children have equal risks for adverse perinatal outcomes (Wen et al., 2012), and, in line with this, recent studies do not stratify for this variable and pool both groups, for instance in (Goisis et al., 2019; Mitter et al., 2022).

We have added this information to the materials and methods section lines 149-153.

6) In the discussion, the effect of IVF embryo culture media from different producers/vendors on the birth weight of IVF children is discussed. The reader should be more clearly warned that the given conclusions are based on only a single study and very clear conclusions cannot be drawn from it yet.

We have further discussed this in the dedicated section on the limitations of the study lines 555-568.

7) The low birth weight of ART children compared to SC children is justified just using a single reference, that indicates the lower birth weight for ART children. However, this topic has been the subject of very active research for decades, and the results have been contradictory so far. Therefore, this topic should be discussed somewhat more openly and emphasizing the inconsistency of the results so far.

We completely agree with the reviewer. The manuscript was originally submitted as a brief report which did not allow for elaborating. We have now added more detail to the introduction on the phenotypes observed in ART children and the factors that have been studied in this regard.

References

- Casteels, K., Ong, K., Philips, D., Bendall, H., Pembrey, M., Poulton, J., and Dunger, D. (1999). Mitochondrial 16189 variant, thinnnes at birth, and type-2 diabetes. *The Lancet* 353, 1499–1500. .
- Ebner, S., Mangge, H., Langhof, H., Halle, M., Siegrist, M., Aigner, E., Paulmichl, K., Paulweber, B., and Datz, C. (2015). Mitochondrial Haplogroup T Is Associated with Obesity in Austrian Juveniles and Adults. 1–13. <https://doi.org/10.1371/journal.pone.0135622>.
- Flaquer, A., Baumbach, C., Kriebel, J., Meitinger, T., Peters, A., Waldenberger, M., Grallert, H., and Strauch, K. (2014). Mitochondrial Genetic Variants Identified to Be Associated with BMI in Adults. *PLoS One* 9. <https://doi.org/10.1371/JOURNAL.PONE.0105116>.
- Fu, J., Ma, M.M., Pang, M., Yang, L., Li, G., Song, J., Zhang, J.W., and Cui, Y. (2019). Broadening the phenotype of m.5703G>A mutation in mitochondrial tRNAAsn gene from mitochondrial myopathy to myoclonic epilepsy with ragged red fibers syndrome. *Chin Med J (Engl)* 132, 865–867. <https://doi.org/10.1097/CM9.000000000000151>.
- Goisis, A., Remes, H., Martikainen, P., Klemetti, R., and Myrskylä, M. (2019). Medically assisted reproduction and birth outcomes: a within-family analysis using Finnish population registers. *The Lancet* 393, 1225–1232. [https://doi.org/10.1016/S0140-6736\(18\)31863-4](https://doi.org/10.1016/S0140-6736(18)31863-4).
- Herrero-Martí-N, M.D., Ayuso, T., Tuñón, M.T., Martín, M.A., Ruiz-Pesini, E., and Montoya, J. (2010). A MELAS/MERRF phenotype associated with the mitochondrial DNA 5521G>A mutation. *J Neurol Neurosurg Psychiatry* 81, 471–472. <https://doi.org/10.1136/jnnp.2009.173831>.
- Jia, W., and Higgs, P.G. (2008). Codon usage in mitochondrial genomes: Distinguishing context-dependent mutation from translational selection. *Mol Biol Evol* 25, 339–351. <https://doi.org/10.1093/molbev/msm259>.
- von Kleist-Retzow, J.-C., Cormier-Daire, V., Viot, G., Goldenberg, A., Mardach, B., Amiel, J., Saada, P., Dumez, Y., Brunelle, F., Saudubray, J.-M., et al. (2003). Antenatal manifestations of mitochondrial respiratory chain deficiency. *J Pediatr* 143, 208–212. [https://doi.org/10.1067/S0022-3476\(03\)00130-6](https://doi.org/10.1067/S0022-3476(03)00130-6).
- Knoll, N., Jarick, I., Volckmar, A.L., Klingenspor, M., Illig, T., Grallert, H., Gieger, C., Wichmann, H.E., Peters, A., Wiegand, S., et al. (2014). Mitochondrial DNA variants in obesity. *PLoS One* 9. <https://doi.org/10.1371/JOURNAL.PONE.0094882>.
- Li, D., Liang, C., Zhang, T., Marley, J.L., Zou, W., Lian, M., and Ji, D. (2022). Pathogenic mitochondrial DNA 3243A>G mutation: From genetics to phenotype. *Front Genet* 13, 1–14. <https://doi.org/10.3389/fgene.2022.951185>.
- Mertens, J., Zambelli, F., Danneels, D., Caljon, B., Sermon, K., and Spits, C. (2019). Detection of Heteroplasmic Variants in the Mitochondrial Genome through Massive Parallel Sequencing. *Bio Protoc* 9, 1–19. <https://doi.org/10.21769/BioProtoc.3283>.
- Mertens, J., Regin, M., De Munck, N., Couvreur de Deckersberg, E., Belva, F., Sermon, K., Tournaye, H., Blockeel, C., Van de Velde, H., and Spits, C. (2022). Mitochondrial DNA variants segregate during human preimplantation development into genetically different cell lineages that are maintained postnatally. *Hum Mol Genet* 31, 3629–3642. <https://doi.org/10.1093/HMG/DDAC059>.
- Mitter, V.R., Fasel, P., Berlin, C., Amylidi-Mohr, S., Mosimann, B., Zwahlen, M., von Wolff, M., and Kohl Schwartz, A.S. (2022). Perinatal outcomes in singletons after fresh IVF/ICSI: results of two cohorts and the birth registry. *Reprod Biomed Online* 44, 689–698. <https://doi.org/10.1016/j.rbmo.2021.12.007>.
- Nardelli, C., Labruna, G., Liguori, R., Mazzaccara, C., Ferrigno, M., Capobianco, V., Pezzuti, M., Castaldo, G., Farinaro, E., Contaldo, F., et al. (2013). Haplogroup T Is an Obesity Risk Factor : Mitochondrial DNA Haplotyping in a Morbid Obese Population from Southern Italy. 2013. .

Nelissen, E.C., Van Montfoort, A.P., Coonen, E., Derhaag, J.G., Geraedts, J.P., Smits, L.J., Land, J.A., Evers, J.L., and Dumoulin, J.C. (2012). Further evidence that culture media affect perinatal outcome: Findings after transfer of fresh and cryopreserved embryos. *Human Reproduction* 27, 1966–1976. <https://doi.org/10.1093/humrep/des145>.

Parker, E., Phillips, D.I.W., Cockington, R.A., Cull, C., and Poulton, J. (2005). A common mitochondrial DNA variant is associated with thinness in mothers and their 20-yr-old offspring. *Am J Physiol Endocrinol Metab* 289. <https://doi.org/10.1152/AJPENDO.00600.2004>.

Schon, E., DiMauro, S., and Hirano, M. (2012). Human mitochondrial DNA: roles of inherited and somatic mutations. *Nat Rev Genet* 13, 878–890. <https://doi.org/10.1038/nrg3275>. Human.

Shtolz, N., and Mishmar, D. (2023). The metazoan landscape of mitochondrial DNA gene order and content is shaped by selection and affects mitochondrial transcription. *Commun Biol* 6, 1–15. <https://doi.org/10.1038/s42003-023-04471-4>.

Wei, W., Tuna, S., Keogh, M.J., Smith, K.R., Aitman, T.J., Beales, P.L., Bennett, D.L., Gale, D.P., Bitner-Glindzicz, M.A.K., Black, G.C., et al. (2019). Germline selection shapes human mitochondrial DNA diversity. *Science* (1979) 364. <https://doi.org/10.1126/SCIENCE.AAU6520>.

Wen, J., Jiang, J., Ding, C., Dai, J., Liu, Y., Xia, Y., Liu, J., and Hu, Z. (2012). Birth defects in children conceived by in vitro fertilization and intracytoplasmic sperm injection: A meta-analysis. *Fertil Steril* 97, 1331-1337.e4. <https://doi.org/10.1016/j.fertnstert.2012.02.053>.

Yang, T.L., Guo, Y., Shen, H., Lei, S.F., Liu, Y.J., Li, J., Liu, Y.Z., Yu, N., Chen, J., Xu, T., et al. (2011). Genetic Association Study of Common Mitochondrial Variants on Body Fat Mass. *PLoS One* 6, 21595. <https://doi.org/10.1371/JOURNAL.PONE.0021595>.

Zambelli, F., Vancampenhout, K., Daneels, D., Brown, D., Mertens, J., Van Dooren, S., Caljon, B., Gianaroli, L., Sermon, K., Voet, T., et al. (2017). Accurate and comprehensive analysis of single nucleotide variants and large deletions of the human mitochondrial genome in DNA and single cells. *European Journal of Human Genetics* 25, 1229–1236. <https://doi.org/10.1038/ejhg.2017.129>.

Zambelli, F., Mertens, J., Dziedzicka, D., Sterckx, J., Markouli, C., Keller, A., Tropel, P., Jung, L., Viville, S., Van de Velde, H., et al. (2018). Random Mutagenesis, Clonal Events, and Embryonic or Somatic Origin Determine the mtDNA Variant Type and Load in Human Pluripotent Stem Cells. *Stem Cell Reports* 11, 102–114. <https://doi.org/10.1016/J.STEMCR.2018.05.007>.

REVIEWER COMMENTS

Reviewer #2 (Remarks to the Author):

I commend the authors on their thorough rebuttal and substantial effort on revisions, which have improved their manuscript. As such, they have addressed the bulk of the reviewer comments. However, several comments remain:

- Lines 295-298: “This suggests that individuals born after ART tend to more frequently carry heteroplasmic variants in the protein coding and/or rRNA regions, in combination with fewer heteroplasmic variants in the other regions compared to their SC peers” – This conclusion drawn from the factor analysis is not supported by the underlying data, since there was no significant differences in the distribution of variants across categories between ART and SC, as reported throughout the first results section (e.g. Fig. 1h-j). The authors therefore should revise this conclusion or provide more evidence or clarification in text to support it.
- Lines 434-436: “These findings suggest that the differences in mtDNA variants between ART and SC children are due to a higher transmission of protein- and rRNA-coding variants by ART mothers” - This conclusion is not supported in the revised text, given no significant difference in transmission was observed between SC vs ART (i.e. Fig. 3a). This is another example of a conclusion drawn from factor analysis that was not supported by analysis of the underlying data. Furthermore, the number of transmitted non-synonymous and rRNA variants is only n=4 in each group (Table S27), with most being synonymous, warranting caution. The authors should thus revise this conclusion (and title of this results section) or provide other evidence to support it.
- The response to the point on using population frequency data (citing Wei et al) was confusing and did not provide clear support. One suggestion is to use frequency data from population databases (e.g. gnomAD, MITOMAP). For example, population frequency <1/50000 in these databases can be used as supporting evidence of pathogenicity in ACMG guidelines (PMID: 32906214). Are low birthweight genotypes enriched in variants meeting this pathogenicity criteria, or are any other population frequency differences observed? This could help support their conclusions.
- Their new assessment of variants across the oocyte donors with multiple tissues showed that 10/24 heteroplasmic variants were tissue-specific, suggesting they may not be present in the germline (and thus unlikely to contribute to birthweight effects). Since many of their conclusions are drawn from analysis of blood or saliva samples, this should be mentioned as another possible limitation of this study in the discussion.

- Tables S14/S15 report total n=388 SC individuals used for calculations, but only 164 are listed in the text (i.e. line 305), also per Table 1. Can the authors clarify?

Reviewer #2 comments on authors response to Reviewer #1:

The authors have satisfactorily responded to reviewer #1's points, with one note.

For point 2: "Although the inclusion of categorical variables in an LDA approach is somewhat against the basic assumption of multivariate normality, as long as the data projected on the line through both group medians follow a bi-variate normal distribution, the discriminant analysis procedure is sufficiently robust"

– The authors should show that their data follows this assumption, and or repeat their analysis excluding the categorical variables. If significance relies on inclusion of categorical variables, then this should be mentioned in the text as a caveat of their results.

Reviewer #3 (Remarks to the Author):

All questions of the Reviewer have been adequately addressed, and the new analyses add additional value to the manuscript.

Dear reviewers,

Thank you for your assessment of the revised version of our manuscript and for the thorough evaluation of our rebuttal. We understand some questions remained unanswered, which we hope you find now satisfactorily addressed in this second revision.

We have made changes to the text (highlighted in orange), corrected a mistake in figure 2, adjusted figure 3 to the request of reviewer 2, added a supplementary table with the potential pathogenicity of the non-synonymous variants and have created a supplementary figure to clarify the factor analysis.

We hope you agree that these changes have further strengthened the manuscript and find it ready for publication.

Looking forward to hearing from you.

Point-by-point rebuttal:

- Lines 295-298: “This suggests that individuals born after ART tend to more frequently carry heteroplasmic variants in the protein coding and/or rRNA regions, in combination with fewer heteroplasmic variants in the other regions compared to their SC peers” – This conclusion drawn from the factor analysis is not supported by the underlying data, since there was no significant differences in the distribution of variants across categories between ART and SC, as reported throughout the first results section (e.g. Fig. 1h-j). The authors therefore should revise this conclusion or provide more evidence or clarification in text to support it.

The reviewer is correct to state that there are no significant differences in each category of variants on their own. The interesting and added value of the factor analysis is that it considers each of the categories for each sample, thus extracting a pattern not observed in each category on its own.

To further clarify how this is achieved, we have created an additional figure that we have added to the supplementary methods.

The figure illustrates the relationship between the heteroplasmic loads of the variants in each of the regions and the resulting factor, for all individuals in this study. The figure shows as an example factor 2. Each individual is represented by a triad of dots. The purple dot is their factor 2 score. The blue dot represents the sum of the heteroplasmic loads of the HV, NonCod, TAS and OHR, in per unit (instead of percent). The green dots represent the sum of loads for Syn, NonSyn and rRNA variants. Since the loads in the HV, NonCod, TAS and OHR negatively influence the value of factor 2, they have been put in the negative, and the loads in Syn, NonSyn and rRNA in the positive as they positively impact the factor score. The data has been split in quartiles based on their factor 2 score rank, to help visualizing the differences between

SC and ART individuals in two populations of different size. The figure visualizes the tug between the categories positively and negatively affecting the factor score, and shows the values for each of these categories together for each individual, instead of separately as shown in figure 1.

- Lines 434-436: “These findings suggest that the differences in mtDNA variants between ART and SC children are due to a higher transmission of protein- and rRNA-coding variants by ART mothers” - This conclusion is not supported in the revised text, given no significant difference in transmission was observed between SC vs ART (i.e. Fig. 3a). This is another example of a conclusion drawn from factor analysis that was not supported by analysis of the underlying data. Furthermore, the number of transmitted non-synonymous and rRNA variants is only n=4 in each group (Table S27), with most being synonymous, warranting caution. The authors should thus revise this conclusion (and title of this results section) or provide other evidence to support it.

We agree that the sample size of transmitted variants is low and precludes robust conclusions on this aspect. We have deleted the factor analysis in this part, both from the figure and from the text, and adjusted this conclusion and title of the results section, which now reads ‘ART individuals more frequently show de novo non-synonymous variants than their spontaneously conceived peers’.

- The response to the point on using population frequency data (citing Wei et al) was confusing and did not provide clear support. One suggestion is to use frequency data from population databases (e.g. gnomAD, MITOMAP). For example, population frequency <1/50000 in these databases can be used as supporting evidence of pathogenicity in ACMG guidelines (PMID: 32906214). Are low birthweight genotypes enriched in variants meeting this pathogenicity criteria, or are any other population frequency differences observed? This could help support their conclusions.

Thank you for clarifying this question, for the concrete proposal and the especially useful reference. We have queried MITOMAP for all the non-synonymous variants included in the birthweight analysis. We find that 46 of the 80 variants (57.5%) have reported frequencies <0.002%, which is significantly higher than the total of MITOMAP variants that are under this threshold (19%) (PMID: 32906214).

Mutpred2-based scoring of the non-synonymous changes classified 15 of the 80 variants as potentially pathogenic (this did not include the 11 indels, which are not scored by this algorithm). Combining both scores yields 50 variants that are potentially pathogenic. Taken together, the results suggest that for 62.5% of the identified non-synonymous variants there is evidence to support categorizing them as potentially pathogenic.

We tested if there was a difference in the incidence of these potentially pathogenic variants between birthweight groups, and found, similarly as when computing for all non-synonymous variants, that individuals born with a birthweight under P25 had a higher incidence of these variants than those born with a birthweight over P25 (14% in >P25, 21% in <P25), but statistical significance was lost, likely due to the reduction in sample size.

We have included the data as a supplemental table S24, and explained this in the text:

“As in most cases, the non-synonymous variants were not known to be associated with mitochondrial disease, we categorized them on whether there was further supporting evidence for their potential pathogenicity. Fifteen variants (18.75%) had a MutPred2 score over 0.61 (Pejaver et al., 2017) and forty-six variants (57.7%) had a reported frequency under 0.002 in MITOMAP (mitomap.org, (Lott et al., 2013)), a threshold that has been proposed as supporting evidence for pathogenicity (McCormick et al., 2020). This incidence of uncommon variants in our population is remarkably higher than that found in the MITOMAP database,

where 19% of variants meet this threshold (McCormick et al., 2020). Taking the MutPred2 scores and the MITOMAP frequencies together, 50 of the 80 identified non-synonymous variants have supporting evidence for being potentially pathogenic (62.5%, TableS24). Comparison of their incidence in individuals under and above P25 yielded no statistically significant differences, likely due to the limited sample size (21% in <P25, 14% in >P25, $p=0.270$), and we did not further factor this subdivision of the variants in the subsequent analysis.”

- Their new assessment of variants across the oocyte donors with multiple tissues showed that 10/24 heteroplasmic variants were tissue-specific, suggesting they may not be present in the germline (and thus unlikely to contribute to birthweight effects). Since many of their conclusions are drawn from analysis of blood or saliva samples, this should be mentioned as another possible limitation of this study in the discussion.

We agree that these variants are possibly not present in the germline of the donor and are therefore not transmitted, but on the other hand, we cannot establish if they are tissue specific, or if they appeared in a somatic lineage early in development and would be present in other tissues.

We have added this point to the limitations of the study “Finally, it is important to consider that when analyzing only one tissue per individual, it cannot be excluded that the identified heteroplasmic variants are tissue-specific, in which case they had a limited potential to affect the individual’s birthweight.”

- Tables S14/S15 report total $n=388$ SC individuals used for calculations, but only 164 are listed in the text (i.e. line 305), also per Table 1. Can the authors clarify?

Thank you for spotting this mistake, we have adjusted tables S14 and S15.

Reviewer #2 comments on authors response to Reviewer #1:

The authors have satisfactorily responded to reviewer #1’s points, with one note.

For point 2: “Although the inclusion of categorical variables in an LDA approach is somewhat against the basic assumption of multivariate normality, as long as the data projected on the line through both group medians follow a bi-variate normal distribution, the discriminant analysis procedure is sufficiently robust”

– The authors should show that their data follows this assumption, and or repeat their analysis excluding the categorical variables. If significance relies on inclusion of categorical variables, then this should be mentioned in the text as a caveat of their results.

Given that most variables in the models are categorical, excluding them is not an option. To check the robustness of the performed LDA in the presence of categorical predictors, we analyzed the distribution of the discriminant scores. The discriminant scores should reflect a bi-normal distribution as it is the result of a linear combination of the predictors used which are assumed mutually normally distributed.

The distribution of the LDA scores for the P10 model reflects a bivariate normal distribution. This is in line with the Wilk’s Lambda test with a $p<0.001$, suggesting this is a robust model despite the fact that it is built on a majority of categorical variables.

Simple Dot Plot of Discriminant Scores from Function 1 for Analysis P10

Simple Dot Plot of Discriminant Scores from Function 1 for Analysis P25

For the discriminant scores of the second model, the separation between the two groups is smaller such that both modes are in each other's proximity which merges the bi-normal distribution into a unimodal distribution. This is also consistent with the Wilk's Lambda test indicating that the model is just significant, with a $p=0.046$.

Overall, this is consistent with the two model's parameters shown in Table 3 and Table 4, where it is clear that the model for P10 is better at correctly predicting the birthweight of the individuals. The properties of the LDA scores show that the models are sufficiently robust considering the presence of the categorical variables. This conclusion is also supported by the good cross validation which reveals that the discriminators could be applied for individuals that were not included in the training set. Nonetheless, there is clearly room for improvement in the P25 model, which future prospective studies should certainly address. We have included a caveat in the limitations of the study: **“Moreover, while the model to discriminate the 10th birthweight percentile is robust, the model for the 25th birthweight percentile is only just significant, and will require further adjustment and improvement.”**

REVIEWERS' COMMENTS

Reviewer #2 (Remarks to the Author):

The authors have satisfactorily addressed the remaining reviewer comments.